# Inferring the internal structure of groups through the integration of statistical learning and causal reasoning

Isaac Davis [1] ✉, Julian Jara-Ettinger [1,2] & Yarrow Dunham [1,2]

Human social life unfolds within richly structured networks of overlapping relationships, including friendships, hierarchies, and collaborations. Yet the observable interactions that reveal these networks are often sparse and noisy, making it unclear how people could infer the latent structure of their social environments from such limited evidence. We propose that humans integrate domain-general statistical learning with domain-specific models of social structures to rapidly construct causal representations that support explanation, prediction, and planning. Across three behavioral experiments, we show that participants can infer underlying social structures (Experiment 1), predict social behavior (Experiment 2), and reason about the spread of social influence (Experiment 3), based on brief, abstract videos of social interactions. These judgments were closely captured by a computational model grounded in our account and could not be explained by simpler cue-based accounts. Statistical learning and causal reasoning operate in concert to support rapid, flexible understanding of social structures.

Human sociality is remarkably varied and complex. Our everyday experiences unfold within a web of interconnected social structures, including family and friendship networks, professional institutions, and cooperative alliances. These structures vary widely in scale, complexity, and duration, and each encode a distinct set of normative implications for behavior. This complexity creates a formidable learning challenge: faced with sparse and noisy data concerning how individuals interact, how do we infer these structures and their central entailments?

Consider, for example, the first day at a new office job. While some rules and roles will be explicitly transmitted, much must be inferred. Who wields power and influence? Are there factions and rivalries? To whom should you turn for information or advice? To complicate matters further, these relations rarely exist in isolation and must be distinguished from other overlapping structures such as friendship cliques, cultural commonalities, and so on.

Understanding these aspects of our surroundings is critical for embedding ourselves into social environments, from large-scale institutions like governments and corporations to more transient organizations like sports teams or friend groups. While a wealth of research focuses on the early emerging tendency to represent social environments in terms of groups, even on the basis of minimal evidence, most of this work focuses on intergroup dynamics: to what group do I belong, what general features characterize members of groups, and what characterizes interactions across group boundaries[1-4]? At this level of analysis, groups are loose alliance structures in which members are largely substitutable: what matters is membership. But this emphasis can obscure vital aspects of intragroup structure. Hierarchical relations within groups, as well as patterns of internal friendship and alliance, are common across species and central to survival and reproductive success[5-7]. As illustrated by the opening example, however, human social structures are uniquely complex, involving multiple overlapping social dynamics operating at multiple scales and admitting of substantial cultural variation[8-10]. The generativity and flexibility of human social structures make it clear that our capacity to recognize social structure cannot be fully "pre-programmed:" we must be able to learn and reason about novel social structures in novel social environments.

How do we achieve this? While some group structures are explicitly taught (e.g., the structure of governments), people often

[1]Department of Psychology, Yale University, New Haven, CT, USA. [2]Wu Tsai Institute, Yale University, New Haven, CT, USA. ✉e-mail: isaac.davis@yale.edu

encounter novel environments where the latent social structures are unknown. In these cases, the ability to reconstruct the underlying internal structure is critical for making sense of the observed interactions, predicting individual and group-level behavior, and knowing how to exert influence and succeed within the social collective.

Prior work on social network cognition suggests that people infer social structures through the lens of cognitive "schemas"[11–13]—representations of common social network structures that shape how people perceive and remember collections of dyadic relations between individuals (often in a way that leads to inaccurate representations of network structure, e.g., refs. 14,15). Traditionally, these social schemas are thought of as simplified representations tied to specific heuristics or expectations. For example, one schema might represent friendship cliques, tied to a transitivity heuristic entailing that two people who share a mutual friend are likely also friends[16,17]; another schema might represent hierarchies, tied to a linear-order heuristic entailing that everyone in the group falls into a single "pecking order"[18]. Our approach diverges in the assumed specificity and flexibility of these representations: rather than interpreting social data through one of a fixed set of specific schemas, we posit that people can flexibly compose basic representations to reason about more complex structures that don't necessarily map onto one existing schema. For example, we might recognize that a high-level corporate hierarchy (e.g., management oversees both sales and research) also contains several internal hierarchies (within each of sales and research), which may also overlap with other non-hierarchical structures within the group (e.g., some salespeople are in shared friend groups with some researchers). Thus, our framework proposes a richer and more flexible compositional representation space than what schemas support. That said, schemas also likely contribute to how people infer and reason about latent social structures. More precisely specifying their unique contribution alongside the more compositional processes we propose here will be an important topic for future work.

To achieve these flexible inferences in a tractable way, we propose that this hallmark of human social intelligence involves the integration of two capacities. The first is a domain-general statistical learning mechanism over a space of abstract data structures (e.g., categories, hierarchies, networks, etc.), which can be composed into representations of arbitrary complexity[19,20]. This capacity to abstract discrete events into structured statistical summaries (i.e., statistical reasoning) is at work from early in infancy[21,22] and appears to be a domain-general capacity that applies to many modalities including vision[23,24], language[25,26], and social reasoning, such as how we cluster individuals into groups by perceived similarity[27–29]. In a social context, this allows us to seamlessly represent an observed sequence of pairwise interactions as an abstracted collection of relations between individuals. To this end, we hypothesize that people have a space of representational primitives that can be composed into structures of arbitrary complexity[30–32], where social subgroups and roles are formalized as nodes, and relations between subgroups are formalized as edges. Figure 1a shows three examples of such representations which reflect different types of social relations but share the same groupings. The flexibility afforded by this approach is crucial, as there are usually a vast array of plausible latent structures, and the complexity of these structures is rarely known a priori. This representational space is instantiated as a generative grammar—i.e., a grammar of social structures—containing a set of basic representations and a set of rules for combining them into more complex representations[10,33]. However, this flexibility also poses a serious challenge: how can an observer search through an infinite space of possible social structures to identify a small handful that are most likely, given the observed behavior?

We addresses this challenge by using a non-parametric "Chinese Restaurant Process" (CRP) to efficiently navigate through this infinite search space[34,35]. Intuitively, a CRP starts by positing an extremely

simple structure (e.g., a single cluster), then gradually proposes incremental changes as new agents are either placed in an existing cluster (with probability proportional to the size of that cluster) or in a new cluster that becomes available for subsequent rounds of clustering. This allows the observer to infer the appropriate level of complexity necessary to explain the data. Prior work has used CRP-based clustering methods to model social structure inference[28,29], but this work focused on clustering agents by perceived similarity over a static set of features, and did not model inter-cluster relations. Our approach modifies the standard CRP by clustering agents based on similar patterns of interaction with other agents, and adds a separate step for inferring edges between clusters to represent social relations. By incrementally composing these representational primitives via non-parametric inference, an observer can efficiently construct structures with an appropriate level of complexity for explaining a set of observations. While our framework is posed at a computational level of analysis[36], this CRP process also offers an initial hypothesis about the algorithmic implementation of how people infer social structures.

On their own, these abstract graph representations are not enough to support social inference and prediction, as the way we interpret an interaction between agents will depend on the specific content of that interaction (e.g., a work order, an advice request, or a social invitation). Thus, these representations must also include specific content about the nature of different social relations, and how those relations causally influence social interactions. Thus, the second capacity we posit is a domain-specific intuitive theory of social structures[37–40], which encodes expectations about the kinds of relations that exist between social agents (e.g., authority, friendship), how these relations are organized within social collectives (e.g., power hierarchies, friendship cliques), and how they causally influence social interactions (e.g., who tends to give orders whom?; who tends to share compliments with whom?). We refer to these intuitive theories as naive sociologies, and formalize them as causal models that link structured, graph-like representations of social groups to observed behavior, by positing learned and culturally-variable expectations about social interactions. For example, an authority relation between Susan and Bob might imply that Susan will tell Bob what to do more often than vice versa, whereas a friendship relation between Susan and Bob might imply more symmetric interactions (e.g., inviting each other to socialize). Furthermore, these intuitive social expectations can themselves be composed along with the corresponding structure representations: for example, the basic expectations associated with a friendship clique can be composed with the basic expectations associated with a hierarchy to obtain a set of expectations governing a hierarchy within a friendship clique.

While an exhaustive characterization of naive sociology is beyond the scope of this paper, we hypothesize that, at a minimum, it must contain expectations about three phases of a social interaction in order to support the kinds of inferences we explore. These are: who is likely to initiate an interaction (e.g., a manager is more likely to initiate an order than a low-level employee), who is the likely target of an interaction (e.g., orders tend to flow from superiors from subordinates), and how is the recipient likely to respond (e.g., an agent is more likely to fulfill an order coming from a direct superior, but less likely to fulfill the same order coming from a peer or subordinate). Our work assumes the presence of naive sociology in our adult participants. This assumption is reasonable given past work providing compelling evidence that many core building blocks of naive sociology emerge as early as infancy, including a representation of dominance relations[41–44], notions of social affiliation[45,46] and support[2,47], inter-agent competence representations[48], and a general capacity to recognize people's goals[49,50]. What's more, these sociologies exert profound influences across the lifespan, structuring memory, expectations, and myriad social behaviors throughout life[37,51]. For the present studies, we chose to model three social relations. The first two are authority and social

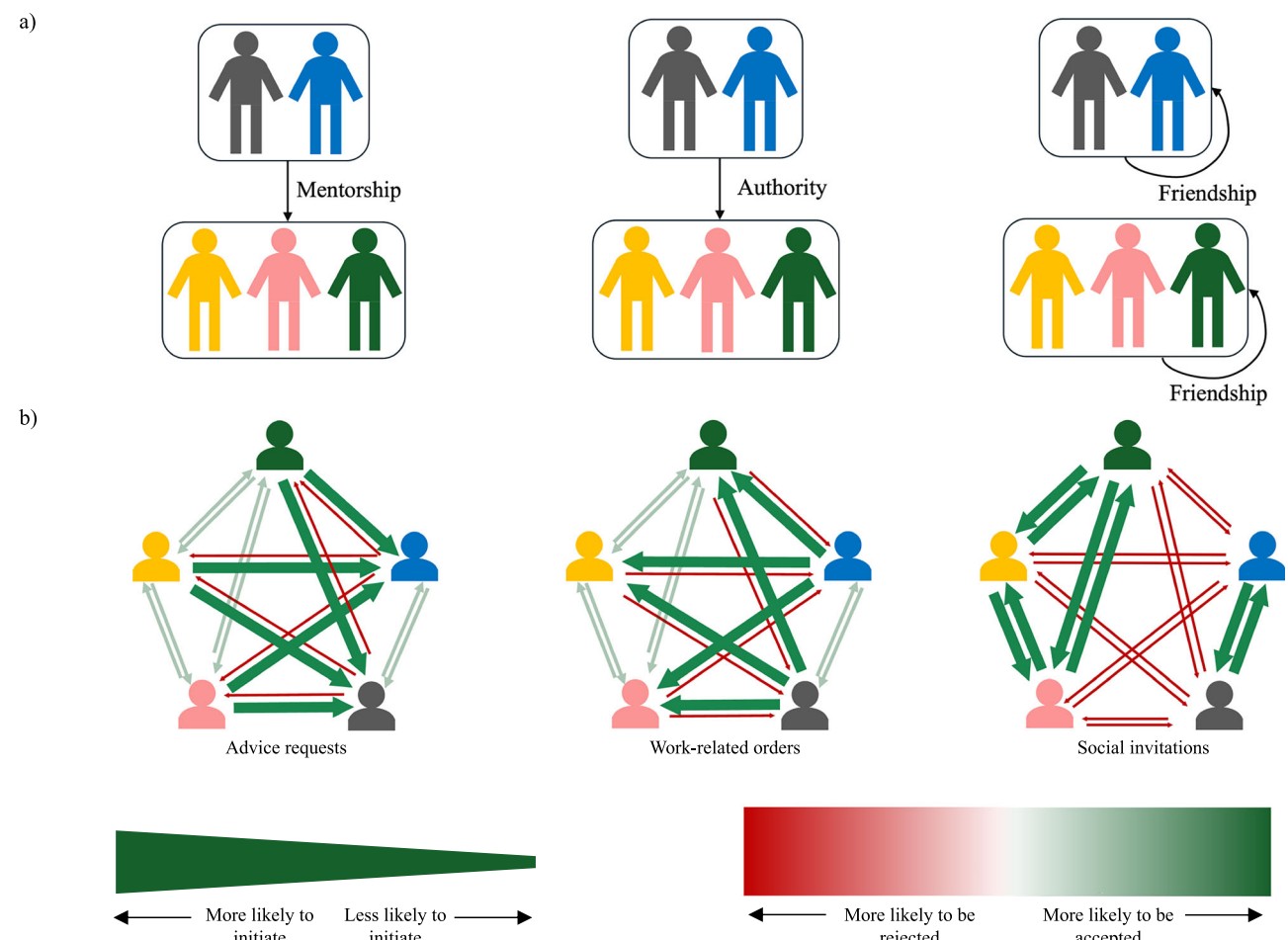

**Fig. 1 | Examples of social structure representations and the generative models associated with each structure. a** Three social structure graphs, each with the same clusters but encoding a different social dynamic. From left to right, these are: mentorship, where arrows go from mentors to mentees; authority, where arrows go from superiors to subordinates; and friendship, where arrows are self-directed to indicate cliques. **b** The expected distribution of interactions encoded by the generative model associated with each structure. Arrows indicate direction of interaction (initiator at the head, recipient at the tail). Arrow width encodes the likelihood of that interaction being initiated, with wider arrows representing more likely interactions. Arrow color encodes the likely response for that recipient: darker green indicates high likelihood of acceptance (e.g., fulfilling a work order or accepting an invitation), while darker red indicates high likelihood of rejection.

closeness, both of which are well-studied and known to be especially salient and early-emerging in children's social representations. However, we also wished to test how well our approach can generalize beyond familiar literatures. To this end, the third social relation we model is mentorship, which is less well-studied and likely to be less salient in children's social development. We therefore formalized and tested naive sociology models of these three social relations, each encoding a set of assumptions about how two agents' relative positions in a social structure affect the likelihood of those agents interacting in a particular way. Crucially, these models encode general, universal expectations (e.g., that orders tend to flow from high-authority to low-authority agents), but also allow for individual or cross-cultural differences in the strength of these expectations in the form of tuneable parameters (e.g., the social cost of refusing an order from a superior agent), which we estimated from preliminary pilot studies (see Supplementary Information). We provide brief, intuitive descriptions of these models in Table 1. A full specification of these models and parameters is provided in Supplementary Information.

Given the set of social relations encoded in a social structure representation, the associated naive sociologies entail a probability distribution over expected interactions, three examples of which are illustrated in Fig. 1b. Because a wealth of research shows that humans' high-level social inferences are well captured by Bayesian inference[52–54], we posit that people treat observed interactions as

stemming from a latent probability distribution that they invert via Bayesian inference to reconstruct the latent social structure that best explains the observed interactions. Our computational model formalizes this proposal, allowing us to make precise and graded quantitative predictions about how people should reason about social structures if our account were correct. By virtue of being a normative model, its alignment to human judgments also reveals people's skill at these inferences.

Here, we present results from three sets of behavioral experiments in which we tested people's capacity to infer latent social structures, use them to predict behavior, and how individual agents might be influenced. Our results reveal that (a) people make high-fidelity reconstructions of the internal structure of a group, all from sparse and noisy social data, (b) people use these representations to judge what might happen next (including after some agents leave the scene) and how members can be influenced, and (c) these graded quantitative judgments reflect both domain-general forms of statistical structure learning and domain-specific social intuitions, integrated in a fashion that tracks with the predictions of a Bayesian inference model.

## Results
Our experiments examined whether observers can infer (potentially complex) latent social structures from a modest number of pairwise

**Table 1 | High-level descriptions of the three naive sociology models used in our studies**

| Sociology (Interaction type) | Interaction term (probability) | | |
| --- | --- | --- | --- |
| | Initiator P(init\|struct) | Recipient P(recip\|init) | Response (resp\|init, recip) |
| Authority (Orders) | Agents with more subordinates more likely to initiate | Agents subordinate to initiator more likely to receive Agents closer to initiator more likely to receive | Agents subordinate to initiator more likely to accept |
| Friendship (Invites) | Agents with more friends more likely to initiate | Agents in same clique as initiator more likely to receive | Agents in same clique as initiator more likely to accept |
| Friendship (Requests) | Agents with more mentors more likely to initiate Agents with more mentees less likely to initiate | Mentors & peers of initiator more likely to receive Agents closer to initiator more likely to receive | Mentors & peers of initiator more likely to accept |

(see Supplementary Information for equations). Each row corresponds to one of the three sociology/interaction types (from top to bottom: authority/orders, friendship/invites, mentorship/advice requests), and each column corresponds to one of the three phases of interaction represented in our model. Within each cell, we list the relevant features of the social structure and how they influence the likelihood of different interactions.

interactions between agents occupying unknown positions within these structures. To this end, we ran three sets of studies in which participants observed short, animated videos of social interactions between a handful of agents, then made judgments about underlying social structures (Experiment 1), future social behavior following a change to the structure (Experiment 2), or the spread of social influence (Experiment 3) among agents in the animation. Across all three experiments, we deliberately chose stimuli that were too sparse and/or noisy to enable straightforward, deductive inferences about the underlying social structures. Furthermore, while participants in Experiments 1 and 2 made judgments about only one type of interaction/social relation at a time (e.g., seeing work-related orders and inferring the associated hierarchies), the stimuli in Experiment 3 each depicted multiple types of interaction, and required participants to reason about different types of (potentially overlapping) social structures simultaneously. Thus, rather than evaluating participants' ability to correctly recover some "ground-truth" social structure, our aim was to characterize participants' graded patterns of uncertainty when making these judgments. Furthermore, because our model embeds a particular set of psychological assumptions about how observers form beliefs about social structure, we also compared the performance of our main model against a panel of alternative accounts, including both non-representational inferences (i.e., making judgments purely based on the surface statistics of the interactions, without drawing deeper inferences about latent social structure) and alternate representational inferences (e.g., drawing inferences about underlying social structure, but leveraging alternative intuitions about social behavior to do so).

## Experiment 1

Note that a pilot version of Experiment 1 previously appeared as a preprint in ref.[55]; that early version involved a smaller sample size ($N = 80$ versus $N = 433$ in the present version), a reduced number of sociological model domains, and a reduced number of alternate models. Experiment 1 comprised three sub-studies exploring participants' ability to make direct inferences about latent social structure from observed interactions between agents. That is, how well can participants recover the underlying set of social arrangements that constrained the observed interactions? Each sub-study (1a, 1b, and 1c) focused on a different type of social interaction, reflecting a different social dynamic (1a: orders/authority; 1b: invitations/friendship; 1c: advice requests/mentorship). Participants first read a set of instructions explaining how to interpret the structure diagrams, the relations encoded by the arrows, and the animated interactions between agents. Participants were then given two chances to correctly answer three comprehension check questions, ensuring a correct interpretation of instructions. Participants who correctly answered all three comprehension checks then watched a series of animated videos, each depicting 8 social interactions between 5

agents. Each interaction depicted one agent (initiator) approach a second agent (recipient), and "communicate" using one of three icons that participants were trained to associate with the three interaction types above. The recipient then responded positively or negatively, indicated by a "thumbs-up" or "thumbs-down." Figure 2a shows examples of our stimulus animations. After each video, participants were shown graphical representations of four candidate social structures of the appropriate type, and used numerical sliders to indicate how likely they thought each one was the underlying structure, given the observations in the animation (see Fig. 2b for an illustration). For each question, we generated predictions using the appropriate main model (i.e., using the "authority" sociology for questions in the hierarchy study) and used the remaining two sociologies as alternate models. Intuitively, these alternate models correspond to interpreting a series of social interactions through a set of assumptions tied to a different type of social interaction (e.g., interpreting a work-related order as if it were a social invitation). These alternate models allow us to evaluate the importance of participants' specific social assumptions when interpreting social data: if participants are largely relying on domain-general statistical learning to make these inferences, then swapping the specific social assumptions should have minimal effect on our model's fit with participant judgments. Note that, in Experiments 2 & 3, we additionally used a non-representational alternate model that simply tracks interaction frequencies between agents. Because this model does not output any explicit structure representations, instead making behavioral predictions from interaction frequencies directly, it cannot be used as an alternate model for Experiment 1, which requires the model to output explicit structure representations.

To select the stimulus animations and candidate social structures for each trial, we first generated a large number of randomly generated interaction sequences, applied our main model to each sequence, and identified the four most likely structures according to the main model. We then chose a set of trials (interaction sequences plus candidate structures) to ensure a range of "degrees of certainty" across trials, including some trials where the main model infers one structure unambiguously, and some trials where the main model identified multiple plausible structures with different degrees of certainty (See "Methods" for more details). This allowed us to determine whether our model can predict both unambiguous judgments, as well as participants' quantitative patterns of uncertainty in more ambiguous trials.

The results of Experiment 1 are summarized in Fig. 3. Across all three sub-studies, our main model strongly and significantly correlated with participant judgments (1a: $R = 0.88$, 95% CI (0.77, 0.94); 1b: $R = 0.89$, 95% CI (0.78, 0.94); 1c: $R = 0.89$, 95% CI (0.79, 0.95)), while alternate models were strongly and significantly anti-correlated with participant responses (i.e., correlations had 95% CIs entirely below 0). Importantly, participant judgments were not "all-or-nothing:" in many

a) Interaction animations

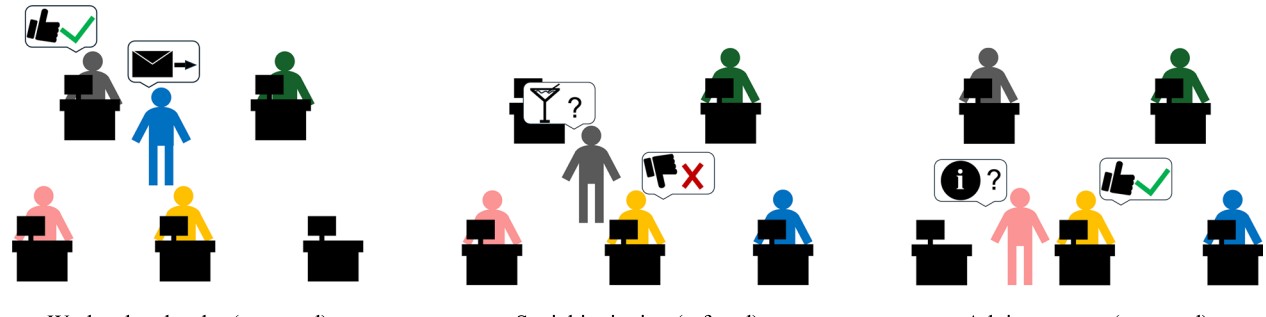

Work-related order (accepted)     Social invitation (refused)     Advice request (accepted)

b) Study 1 response format

Based on the video, which structure seems most likely?

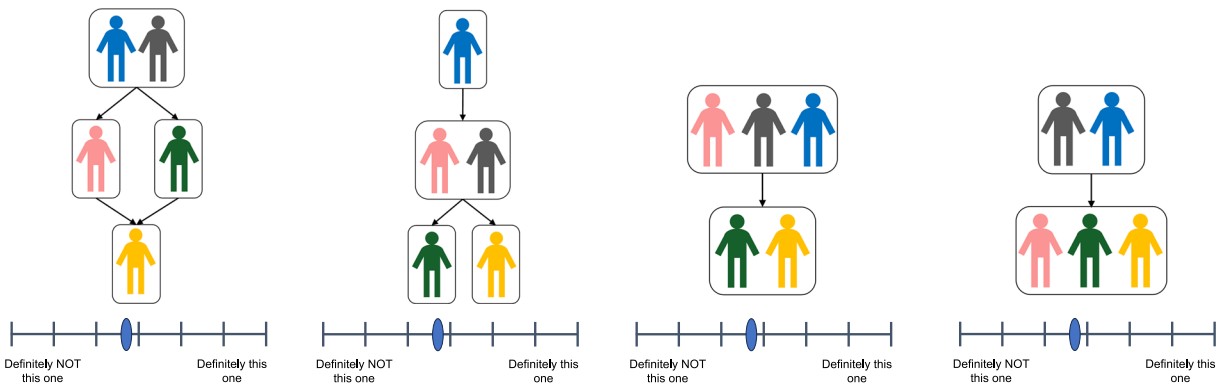

**Fig. 2 | Stimulus and response format for Experiment 1. a** The three interaction types depicted in our stimuli. From left to right: Blue gives Grey a work-related order, which Grey accepts; Grey invites Yellow to socialize after work, which Yellow rejects; Pink gives Yellow a request for advice, which Yellow accepts. **b** An example response format from Study 1a (structure inference: orders/authority).

trials, participants judged multiple candidate structures as plausible to varying degrees, producing a graded range of responses that was also captured by our model. Two examples of such trials are shown in Fig. 4. Thus, even when the data do not allow a straightforward inference about a unique ground-truth social structure, participants can still make graded, quantitative inferences about plausible social structures, and these graded patterns of uncertainty are well captured by our computational model. Additionally, the extremely poor fit with our "alternate sociology" highlights the importance of the observer's specific social assumptions when interpreting social interactions.

## Experiment 2

Experiment 2 comprised three sub-studies exploring participants' predictions about future social interactions, and whether these predictions reflect implicit inferences about the underlying social structures. That is, when predicting new behavior from observed agents, do observers leverage implicit knowledge of the social structure implied by their prior observations? In each trial, participants first watched a sequence of animated interactions similar to those used in Experiment 1. After each video, participants were told that on the next day, one particular agent happened to be out of office, while a second agent wanted to delegate a work-related task (Study 2a), invite someone to hang out after work (Study 2b), or get advice on a work problem (Study 2c). Participants then responded to how likely the second agent was to interact with each other agent using a set of sliders ranging from "Definitely not this person" to "Definitely this person." In order to determine whether participants were implicitly reasoning about social

structure (as opposed to simply extrapolating from observed interaction frequencies), we designed an alternate model that based its predictions purely on interaction frequencies, and chose a set of trials where our main model and frequency model made maximally divergent predictions. We also generated predictions using the two other naive sociology modules for each study, to evaluate whether participants leveraged different expectations depending on the type of interaction being predicted (not pre-registered).

Across all three sub-studies, our main model strongly and significant correlated with participant responses (2a: $R = 0.83$, 95% CI (0.64, 0.92); 2b: $R = 0.92$, 95% CI (0.63, 0.97); 2c: $R = 0.81$, 95% CI (0.60, 0.93)). A summary of these results is shown in Fig. 5. In Study 2a, the frequency model was less strongly correlated than the main model ($R = 0.53$, 95%CI (0.17, 0.77)), though a bootstrapped 95% confidence interval of difference in correlation reveals that this difference is not statistically significant (95% CI: (−0.13, 0.61)), suggesting that a frequency-tracking heuristic can reasonably approximate participant inferences in these trials. In Studies 2b and 2c, our alternate frequency model was anti-correlated with participant responses, suggesting that participants were in fact making implicit inferences about the underlying social structures, rather than relying on surface-level interaction frequencies.

As with Experiment 1, participant judgments were not "all or nothing," and in many cases produced a graded range of inferences, which was captured by our main model but not the frequency model. Figure 6 shows four example trials and results from Experiment 2, illustrating the graded responses produced by participants and the

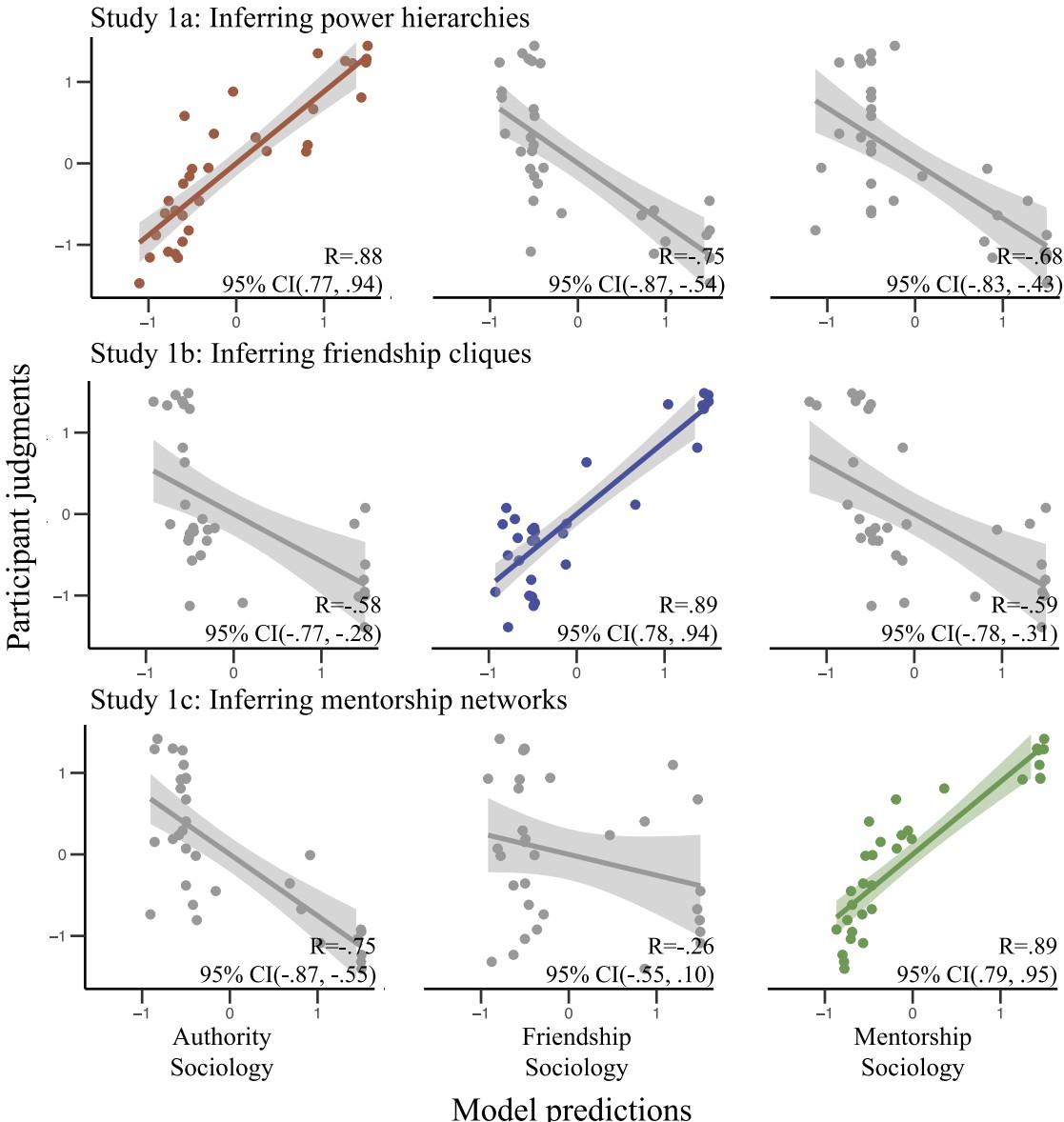

**Fig. 3 | Summary of Experiment 1 results. showing average participant judgments (*y*-axis) against model predictions (*x*-axis).** Each row corresponds to a substudy (from top to bottom: Study 1a, *n* = 89; Study 1b, *n* = 86; Study 1c, *n* = 76), while each column corresponds to a naive sociology module (from left to right: authority, friendship, mentorship). Each point corresponds to the average judgment for a particular structure in a particular trial. Shaded regions indicate 95% confidence intervals around the line of best fit. Scatterplots along the diagonal (highlighted in color) correspond to the "main" models for each study.

main model, but not produced by the frequency model. Importantly, however, when participant judgments were less graded and more clustered (as in study 2b), the main model also produced more clustered predictions, which were not captured by the alternate model. This further supports our hypothesis that participants leveraged implicit inferences about underlying social structure when reasoning about social behavior, even when not explicitly prompted to do so.

The results from the "alternate sociology" models further suggest that participants are leveraging different social expectations when predicting different types of behavior, but also reveal that different aspects of these social expectations can be more or less critical depending on the type of inference being performed. In particular, the most pronounced differences occurred when interactions typically associated with symmetrical relations (i.e., friendship) were interpreted as anti-symmetrical relations (i.e., authority and mentorship), and vice versa. In Study 2a (authority), the "friendship" model

performed significantly worse than the main model ($R = 0.45$, 95% CI (0.05, 0.72)). However, while the "mentorship" model did show a lower correlation than the main model ($R = 0.65$, 95% CI (0.34, 0.84)), a bootstrapped confidence interval over difference in correlation revealed that this difference was not statistically significant (95% CI: ($-0.14$, 0.37)). The results from Study 2c (mentorship) showed a similar pattern: the "friendship" model performed significantly worse than the main model ($R = 0.29$, 95% CI ($-0.13$, 0.62)), but the "authority" model was not significantly different from the main model ($R = 0.85$, 95% CI (0.67, 0.93)). Thus, unlike the structure inference results from Experiment 1, where the mentorship and authority models produced strongly anti-correlated predictions, these results show that the presumed directionality of the relation is much less important when predicting future behavior (though distinguishing symmetrical from anti-symmetrical relations is still critical).

There are several insights raised by these results. First, there is a broad sense in which mentorship and authority carry similar social

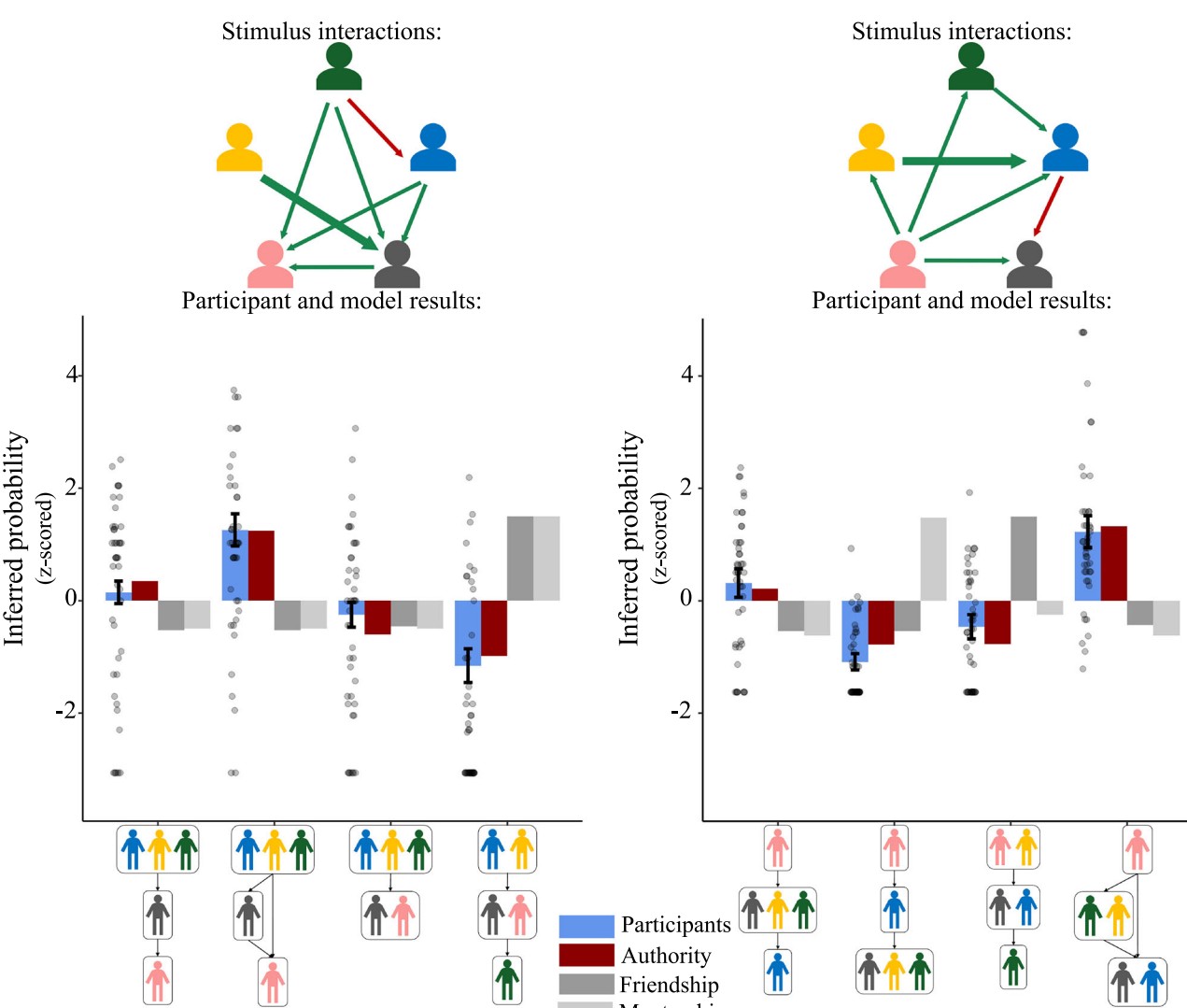

**Fig. 4 | Two example trials from Study 1a (authority), as well as participant responses and all three model predictions for each trial.** Error bars show bootstrapped 95% confidence intervals over participant responses, centered at mean response value. Dots show individual judgments. Interactions for each trial are shown using arrows between agents, with direction indicating initiator (head of arrow) and recipient (tail of arrow), width indicating frequency of interaction (thinner arrows indicate one interaction, thicker arrows indicate two interactions), and color indicating acceptance (green) or rejection (red).

expectations, as both fundamentally involve hierarchies: one based on power (authority) and one based on knowledge (mentorship). Thus, even though the mentorship and authority sociologies are not perfect mirrors of each other (as orders are generally expected to flow down the hierarchy, while advice requests may either flow up to mentors or laterally to peers), the subtle conceptual differences between them were not sufficient to produce strong differences in action predictions. This was further exacerbated by the fact that the trials in this experiment were selected to maximally distinguish between the main sociology model for each study and the non-representational "frequency" model, but not necessarily the alternate sociology models (which were well-distinguished in Experiment 1). There were, however, several individual trials in which the conceptual differences between authority and mentorship were apparent in both participants' and the models' predictions, one of which is highlighted in Fig. 6d. Thus, although the aggregate results show significant similarities between the mentorship and authority model predictions, the differing expectations about interactions between peers is still reflected in trial-level results.

## Experiment 3

The stimuli in the previous experiments each involved only a single type of interaction (and a single type of social structure) at a time. While this enabled a more granular analysis of inferences about specific social relations, real-world social dynamics are obviously much more complex. Even in a small office setting, one would likely observe orders, friendly chit-chat, advice requests, and many other interaction types, even between the same pair of individuals, reflecting multiple (potentially overlapping) social structures within the same group of agents. To better capture this more naturalistic assumption, each stimulus in Experiment 3 depicted all three types of social interaction, and explored participants' expectations about how different types of social structure affect the types of social influence that agents have over each other. Similar to Experiments 1 and 2, each trial showed a sequence of 6 interactions between 5 agents, but unlike previous experiments, each stimulus depicted at least one interaction of each type. After each video, participants were then shown three scenarios in which one agent (A) is considering how to spend their upcoming weekend, and another agent convinces them to do a certain activity.

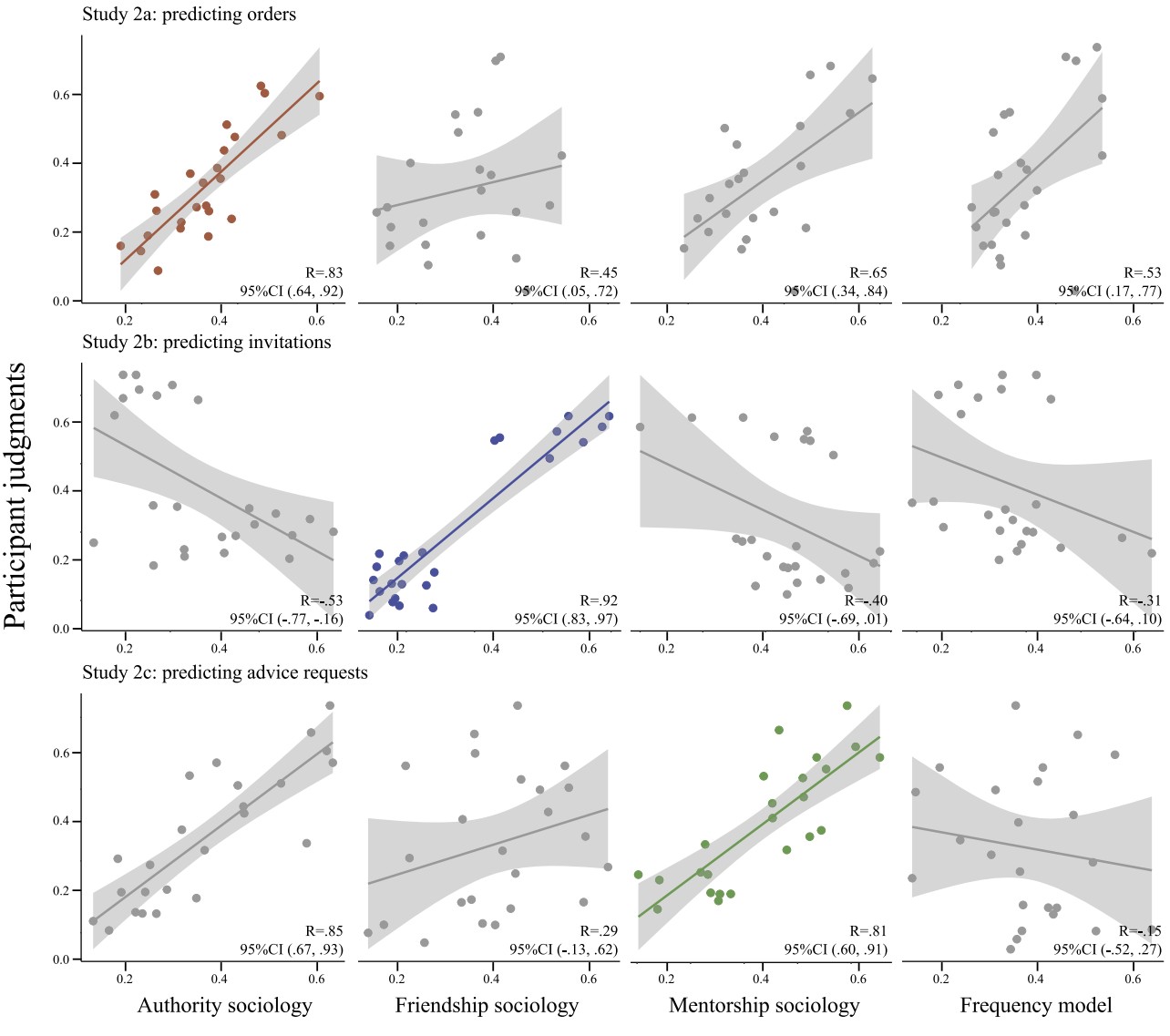

**Fig. 5 | Summary of results from Experiment 2, showing participant judgments (*y*-axis) against model predictions (*x*-axis).** Each point corresponds to the average judgment for a particular agent in a particular trial. Shaded areas indicate 95% confidence intervals around the line of best fit. Plots highlighted in color along the diagonal indicate the main model for each study.

Each row corresponds to a sub-study (from top to bottom: Study 2a, *n* = 87; Study 2b, *n* = 90; Study 2c, *n* = 87), while columns correspond to models (from left to right: authority, friendship, mentorship, and the pure frequency-tracking model).

Participants were then asked which agent convinced A to do the activity. The intuition behind this design is that the type of relationship that two agents have can affect what kind of influence those agents have over each other. For example, we may seek advice from our friends (or those with similar preferences to us) when deciding whether to see a certain movie over the weekend, but seek advice from a professional mentor when deciding whether to take an optional class that teaches a work-related skill. Thus, if participants are making and leveraging implicit inferences about social structure, these inferences should be reflected in their predictions about what kinds of social influence different pairs of agents have over each other. Experiment 3, therefore, investigates (a) whether participants' judgments about social influence reflect implicit inferences about the underlying social structure and (b) whether participants focus on different aspects of the underlying social structure for reasoning about different types of social influence. The three types of scenarios presented in each trial involved an agent considering (a) seeing a certain movie (which we hypothesized to track most closely with friendship relations), (b)

working an extra shift over the weekend (hypothesized to track most closely with authority relations), and (c) taking an optional work-related seminar (hypothesized to track most closely with mentorship relations). Thus, our "main model" for each question type is the sociology module hypothesized to track most closely with the question type. As alternate models, we considered the possibility that participants will attend primarily to one type of social dynamic for all question types, and also the possibility that participants may ignore interaction types and simply assume that agents who interact more frequently have more general influence over each other.

The results of Experiment 3 are summarized in Fig. 7, and results from an example trial are shown in greater detail in Fig. 8. Overall, our main model showed a strong and significant correlation with participant responses for all three question types ("extra shift:" $R = 0.76$, 95% CI (0.36, 0.89); "movie:" $R = 0.95$, 95% CI (0.90, 0.98); "optional class:" $R = 0.82$, 95% CI (0.67, 0.91)). The "single-sociology" alternate models, which assumed that participants would attend primarily to a single social dynamic when predicting social influence, were all uncorrelated

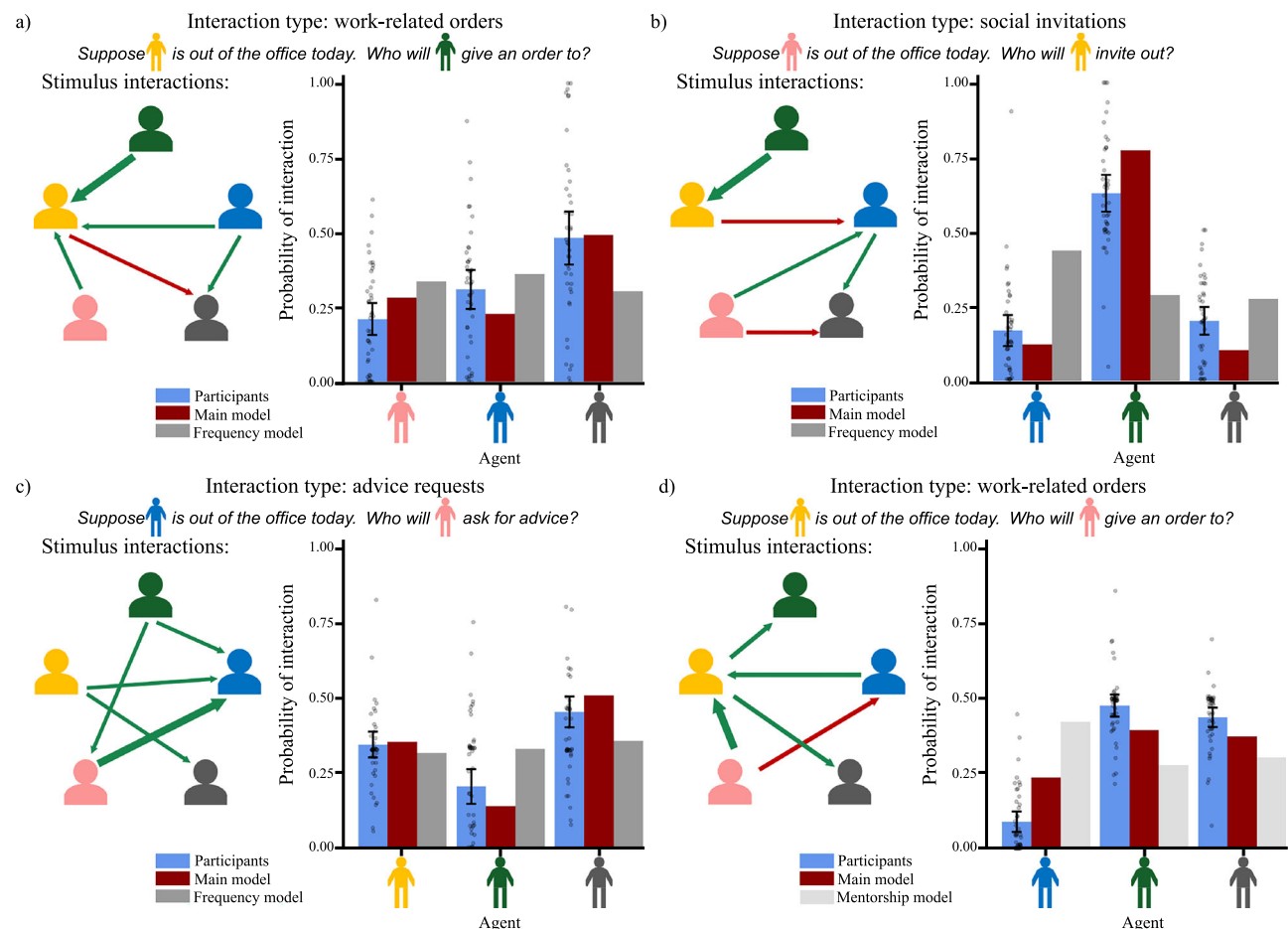

**Fig. 6 | Example stimuli, plus participant responses and model predictions from Experiment 2.** Bars show average probability of interaction judgments (*y*-axis) for each target agent (*x*-axis). Error bars show bootstrapped 95% confidence intervals. Dots represent individual judgments. Interactions for each trial are shown using arrows between agents, with direction indicating initiator (head of arrow) and recipient (tail of arrow), width indicating frequency of interaction (thinner arrows indicate one interaction, thicker arrows indicate two interactions), and color indicating acceptance (green) or rejection (red). **a–c** Example trials from studies 2a (orders), 2b (invitations), and 2c (requests), respectively. **d** An additional trial from Study 2a that highlights the differences between authority and mentorship: both Pink and Blue give orders to Yellow, suggesting they are above Yellow in the hierarchy. When Yellow is out of the office, participants and the authority model predict that Pink will order Green or Grey, both of whom received orders from Yellow. Under the mentorship model, however, the fact that Pink and Blue received advice from Yellow suggests that they are in the same mentorship group, so this model predicts that Pink will most likely interact with Blue. This is contrary to both the authority model and participant judgments, which predict that Pink is least likely to interact with Blue.

or anti-correlated with participant responses. However, the main model outperformed the Frequency model (defined as a bootstrapped 95% CI over difference in correlation entirely above 0) only in the "see a movie" questions: for the "extra shift" and "optional class" questions, the Frequency model did not perform significantly differently than the main model.

These results suggest several conclusions. First, the poor fit of the "fixed sociology" models relative to the main model suggests that participants are implicitly inferring the underlying relational structure, but also selectively attending to specific types of relations when making judgments about social influence. This is further reinforced by the strong quantitative fit between the main model (which used different sociology modules to make predictions for different question types) and participant responses. The relatively strong performance of the Frequency model, which was not significantly different from the main model in the "extra shift" or "optional class" questions, suggests that a simple frequency-tracking heuristic can reasonably approximate participant judgments in these tasks. Given these results, however, it seems unlikely that participants truly ignore implicit information about social structure and simply assume that agents who interact most frequently have the greatest degree of influence over each other: if this were the case, we would expect the main model to perform

similarly to the "fixed sociology" models. That is, if all participants are doing is tracking how frequently agents interact, ignoring the specific types of interactions, then given an even distribution of interaction types across stimuli, we would not expect any one sociology module to perform significantly better than any other. Thus, even if interaction frequencies alone can reasonably approximate participant judgments, it does not seem that participants are only attending to interaction frequencies, nor does it seem that participants weigh each type of interaction equally when reasoning about different types of influence. However, the extreme sparsity of relevant data that participants saw in these trials (in some cases only depicting a single interaction of a certain type) make it impossible to fully infer the underlying social structures, which may force participants to rely in part on the simpler frequency-tracking heuristic.

## Discussion

In order to effectively navigate our everyday social environments, it is crucial for us to recognize the many social dynamics and interconnected social structures underlying those environments. Even a single routine workday might involve dozens of social interactions that require us to track people's roles and relationships within a group. These structures can have a huge influence on our expected social

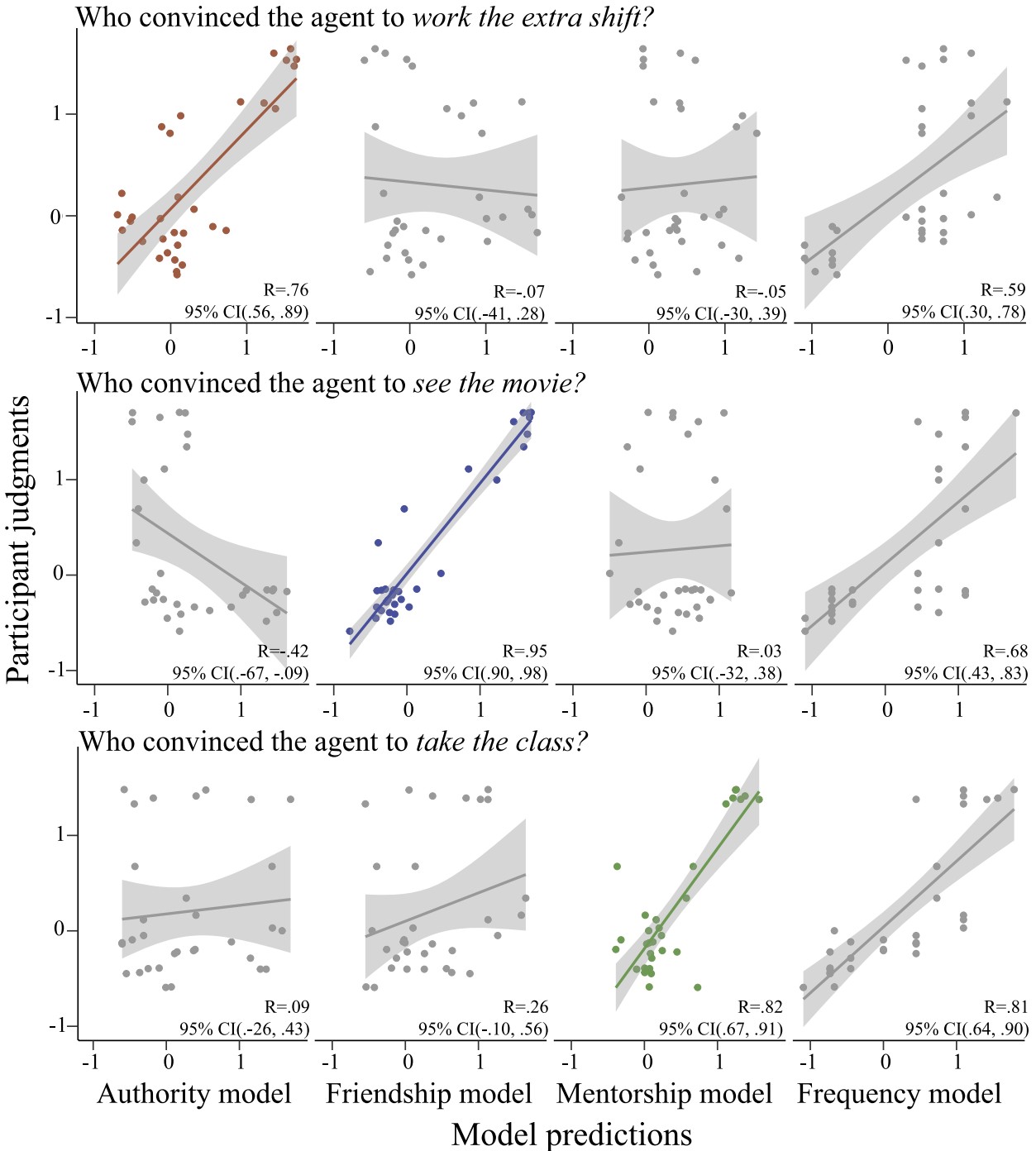

**Fig. 7 | Summary of results from Experiment 3, showing participant judgments (*y*-axis) against model predictions (*x*-axis).** Each point corresponds to the judgment for a single agent in a single trial, averaged across participants (*n* = 87). Shaded areas indicate 95% confidence intervals around the line of best fit. Columns correspond to models, rows correspond to question type ("convinced to worked an extra shift," "convinced to see a movie," "convinced to take a class"). We hypothesized that participant responses to the "extra shift," "movie," and "class" questions would most closely track the predictions of the authority, friendship, and mentorship models, respectively. These "main" models are highlighted in color along the diagonal.

behavior (who can we say no to? who can we ask for advice? who can we confide in?), and failure to accurately detect them can result in serious embarrassment or worse. However, despite a wealth of research exploring how humans carve up the social world into groups and coalitions, much less work has investigated our ability to learn the rich relational structure that exists within groups, and how we can infer this structure in dynamic and noisy social environments.

We proposed that, by integrating a domain-general capacity for statistical structure learning with a domain-specific causal model of

specific social structures, people can make rich inferences about latent social structure from minimal data and leverage these representations to reason about social environments. Our experimental results not only validated this hypothesis, but also demonstrated the remarkable richness of participants' inferences in these tasks. In particular, participants went beyond coarse, categorical judgments about social structures to make finely graded judgments about the relative likelihood of different structures, and these quantitative patterns of uncertainty were accurately captured by our computational model.

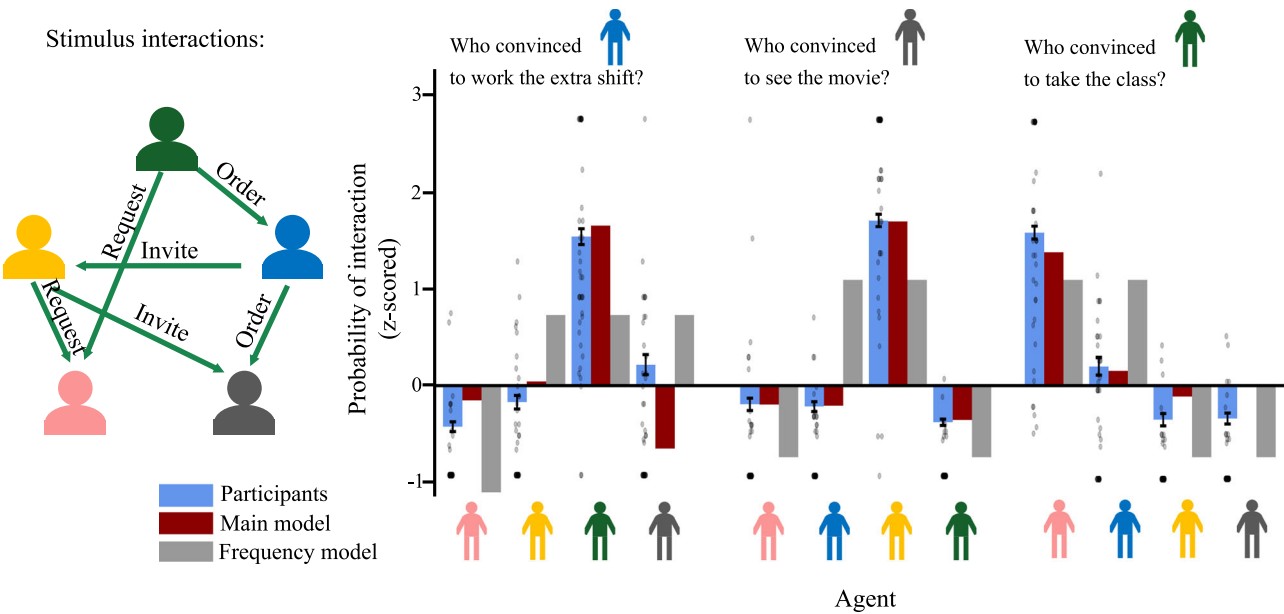

**Fig. 8 | Example trial from Experiment 3, with participant and model predictions.** Error bars show bootstrapped 95% confidence intervals for participant judgments, centered on mean response value. Dots show individual judgments. Interactions for the trial are shown as arrows between agents, with direction indicating initiator (head of arrow) and recipient (tail of arrow), color indicating acceptance (green) or rejection (red), and edge labels indicating type of interaction. Responses shown from all three question types for this trial.

The fact that people could make these fine-grained judgments, even in the face of sparse and noisy data, suggests that this is an extremely important skill for navigating a dynamic social world. This is further supported by the results of our second and third experiments, where participants made implicit inferences about social structure to predict social behavior and the spread of social influence throughout the collective. These results also highlight the importance of leveraging causal representations, which support a wide range of social prediction and planning tasks, as opposed to purely statistical representations, which did not always capture key patterns in participant judgments (as implemented by our "frequency-tracking" alternate model).

The current results establish a basic quantitative capacity to infer complex social structures from sparse and noisy data, opening several lines of future work. First, there are several directions for extending the current framework to explore richer social inferences. This includes inferences based on richer stimuli: while the current experiments used tightly controlled stimuli to ensure that participants attended to the intended set of features encoded in our models, there are many other subtle cues that may influence our interpretation of social data, such as body language[56,57] and other non-verbal signals[58,59]. Future work should therefore investigate how such cues can enable even richer social inferences. Further, even without richer stimuli, people may be able to make more graded inferences about the strength of relationships between specific agents within clusters. Such judgments could be formalized within the current framework: given a set of interactions between agents, one could marginalize the inferred posterior $P(S|D)$ over social structures to compute the probability that two specific agents have a certain relation to one another, which may serve as an index of relationship strength between the two agents. Future experimental work could explore how well the model predicts these more graded judgments among participants.

The current experiments focused on intuitions from US-based participants, but many of these intuitions likely vary across cultures. While nativist notions of early-developing, possibly innate social expectations are often seen as incompatible with the wide cross-cultural variability we observe in social institutions, our core framework is well suited to accommodate both, demonstrating how the two notions can be effectively merged. In particular, our theory posits a (potentially universal) generative grammar over all possible social structures, as well as a set of domain-specific causal models that can vary both qualitatively (e.g., a fundamentally different interpretation of how authority structures influence behavior) and quantitatively (e.g., the expected social cost of refusing an order from a superior) across cultures. By encoding these potential variations as tuneable parameters, and fitting these parameters separately for groups of participants from different cultures, we can use the framework to both estimate cross-cultural differences in basic social expectations, and predict how these differences could produce qualitative differences in more complex social judgments. Exploring these cross-cultural analyses is another important direction for future work.

Finally, an important open question remains as to the origin of our naive sociologies. Are we endowed with innate, perhaps evolutionarily ancient representations of basic social relationships? Developmental evidence reveals that even young infants can recognize and categorize basic social relations, such as authority and closeness[37,40]. How might we combine these basic building blocks to form richer, more complex naive sociologies, and how is this process influenced by social experience and cultural inputs? Recent work has leveraged hierarchical Bayesian models to demonstrate how agents can learn increasingly sophisticated, abstract theories from data collected across multiple domains, though this work has focused primarily on non-social contexts (e.g., ref.[60]). Future work will consider how similar approaches could explain the origin of our naive sociologies and the primitive concepts that enable their development.

Human societies comprise a wide range of complex, overlapping social structures, and our ability to recognize these structures is crucial for embedding ourselves into novel social environments. The present work shows that people can go far beyond coarse group representations based on perceived similarity, and sheds light on how people infer the important relational structures within a group based on patterns of social interaction. Furthermore, while much research in

social cognition has focused on mental representations of other minds, this work highlights how mental representations of social structures can be equally important for navigating our social world, even without explicitly representing the minds of each individual within the structure. Our computational approach suggests that mentalistic and non-mentalistic social representations may be realized within a common framework, as both involve learning causal models of latent states or processes that drive observable behavior. Thus, in addition to exploring how these basic intuitions are learned in the first place, and how variations in these intuitions manifest in more complex social inferences, future work will consider how we flexibly integrate and trade-off between mentalistic and non-mentalistic representations of the social world.

## Methods

### Compliance with ethical regulations

All studies detailed here received ethical approval from the Yale Human Subjects Committee under HSC protocol number 2000020357 and complied with all relevant ethical regulations.

### Models and implementation

We formalize a social interaction $d$ as a 4-tuple that encodes the interaction type ($t$), the initiator, the recipient, and the outcome (i.e., how the recipient responds). Given a set $D = \{d_1, ..., d_n\}$ of observed social interactions between a set of agents, our model outputs a probability distribution $P(S|D)$ over possible social structures, which we compute according to Bayes' rule, i.e.,:

$$P(S|D) \propto P(D|S)P(S) \qquad (1)$$

Here, $P(D|S)$ is the likelihood of observing the interactions in $D$, assuming $S$ is the true structure, and $P(S)$ is the prior probability of structure $S$. Specifications for the prior $P(S)$ and likelihood $P(D|S)$ are provided below.

### Defining social structures and computing the structure prior.

Given a set $A = \{a_1, ..., a_k\}$ of agents and a set $T = \{t_1, ..., t_m\}$ of relation types, we define a social structure over $S$ as an assignment of agents to clusters, representing subgroups and roles, and a set of typed edges between clusters, representing different types of social relations. To compute $P(S)$, we use a CRP[34], implemented via a stick-breaking process[61] with a concentration parameter of 3, to define a prior over all possible partitions of agents into clusters, and for each partition, we assume a 0.5 probability of an edge existing between each pair of groups. Because $P(S)$ specifies a distribution over an infinite set of structures (and is therefore intractable to compute explicitly), we approximated the distribution via a Metropolis–Hastings algorithm[62] iterated for 60,000 samples after a 5000 sample burn-in. In each iteration, we resample the cluster assignments, then resample the edges between clusters.

### Defining naive sociologies and computing the likelihood.

For the likelihood term, we assume each interaction is conditionally independent of all others, given the underlying structure $S$, so that we can factor the likelihood as $P(D|S) = \prod_{i=1}^{n} P(d_i|S)$. Each $P(d_i|S)$ is computed according to a three-stage generative model of pairwise social interaction. The first stage determines which agent will initiate an interaction according to $P_t(init = i|S)$, i.e.,: the probability that agent $i$ will initiate an interaction of type $t$, given the underlying social structure $S$. Next, the initiating agent $i$ chooses which other agent to interact with (the recipient), according to the distribution $P_t(recip = j|init = i, S)$. Finally, the recipient responds to the initiator's interaction either positively or negatively, according to the distribution $P_t(response = yes|init = i, recip = j, S)$. Each term encodes a general intuition about the corresponding stage of a particular type of

interaction, as well as a set of tuneable parameters controlling the strength of those intuitions. Given a set of parameter values for each term, the likelihood of an interaction $P(d|S)$ is equal to the product of the three stage-specific distributions. We fit these parameters to data from an initial pilot study via maximum likelihood estimation (see Supplementary information for a full explanation of model parameters and estimation methods). A high-level explanation of each naive sociology model is provided below:

**Authority/orders.** In the "authority" sociology, orders are more likely to originate from agents with more subordinates, are more likely to be directed to agents subordinate to the initiator, and interactions are more likely to occur between agents that are closer within the hierarchy (e.g., an agent is more likely to give an order to a direct subordinate than a subordinate of a subordinate, etc). Agents are more likely to fulfill orders that originate from superiors than peers or subordinates. Two free parameters $\beta_{down}, \beta_{up} \in (0, 1)$ control the "strictness" of the hierarchy: when $\beta_{down} \approx 1$, agents will only give orders to subordinates directly below them in the hierarchy; when $\beta_{down} \approx 0$, agents are equally likely to give orders to any subordinate, regardless of their distance within the hierarchy. $\beta_{up}$ is an analogous parameter for orders that travel upwards in the hierarchy (which are inherently less likely, but still possible). The third parameter, $\beta_{pos}$, captures the "social cost" of violating a norm by rejecting an order from a legitimate superior: the higher the social cost, the less likely agents are to refuse orders from superiors.

**Friendship/invitations.** In the "friendship" sociology, invitations are more likely to originate from agents in larger friend groups (cliques), are more likely to be directed to agents in the same clique, and are more likely to be accepted by agents in the same clique. A parameter $\beta_{init}$ discounts the rate at which larger friend groups affect the likelihood of agents initiating invitations: for values close to zero, the size of a friend group has little effect on who is likely to initiate an invitation; for large values, invitations are increasingly likely to originate from agents in larger friend groups. Two other parameters, $\beta_{high} > \beta_{low}$, determine the "cliquishness" of the structure: when $\beta_{high}$ is much larger than $\beta_{low}$, invitations will almost always stay within the same clique, and will almost always be rejected when given to an agent from a different clique. When $\beta_{high}$ is closer to $\beta_{low}$, invitations between cliques become more likely.

**Mentorship/advice requests.** In the "mentorship" sociology, advice requests are most likely to originate from agents who have mentors but no mentees, and are least likely to originate from agents with mentees but no mentors. A free parameter $\beta_{request} \in (0, 1)$ affects the strength of this difference: at $\beta_{request} \approx 1$, all agents are equally likely to request advice; at $\beta_{request} \approx 0$, advice requests will only originate from agents with mentors but no mentees. This is similar to a "reverse" version of authority, such that interactions are more likely to be initiated by agents lower in the structure and targeted towards agents higher in the structure (as opposed to orders, which flow from high-authority to low-authority agents). Unlike the authority model, however, agents in a mentorship structure may seek advice from both mentors and peers (i.e., another agent with the same mentor as the initiator), with a parameter $\beta_{recip} \in (0, 1)$ determining the strength of the preference for seeking advice from mentors versus peers. Finally, a "social cost" parameter $\beta_{pos} \in (0, 1)$ determines the likelihood of an agent rejecting a request from a mentee or peer.

**Model predictions.** To generate predictions for each trial, we inputted the interaction sequence depicted in the corresponding stimulus, then approximated $P(S|D)$ via a Metropolis–Hastings algorithm with 60,000 samples and a 5000 sample burn-in as described above. Predictions for Experiment 1 were taken directly from the posterior $P(S|D)$,

while predictions for Experiments 2 and 3 required further manipulation, as described in their respective subsections below. All model code, predictions, data, and analysis scripts are available at https://doi.org/10.17605/OSF.IO/M75DG. In Experiments 2 and 3, we also generated predictions from an alternate model which simply tracks the frequency with which each pair of agents interacted in the stimulus video, then uses this frequency as a direct estimate of the probability that those two agents would interact again (in Experiment 2), or the amount of social influence that those agents have over each other (in Experiment 3).

## Participants

Participants were recruited through Prolific and restricted to those with US-based IP addresses. We recruited 433 participants for Experiment 1, 379 for Experiment 2, and 130 for Experiment 3. Note that participants were evenly divided between three conditions for Experiments 1 & 2, while all participants were assigned to the same condition in Experiment 3. After excluding participants who failed one or more attention checks (see Procedure), our final samples were $n = 251$ for Experiment 1 (51% female, mean age = 37.53, range=18–78), $n = 264$ for Experiment 2 (40% female, mean age = 34.49, range = 18–67), and $n = 87$ for Experiment 3 (53% female, mean age = 42.94, range = 20–73). No statistical methods were used to pre-determine sample sizes, but our target sample sizes were pre-registered and similar to those reported in previous publications on social structure inference through online platforms[27,63]. All participants gave written informed consent and were compensated for their time.

## Stimuli

Each trial in each study consisted of a 20–30 s animated video depicting 6–8 social interactions between agents in a group of 5. Each interaction depicted one agent approaching another, then initiating one of three types of interaction, depicted as a speech bubble with one of three icons. The receiving agent then responded with a second speech bubble depicting a red "X" and "thumbs-down" icon (indicating a "no" response), or a green "checkmark" and "thumbs-up" icon (indicating a "yes" response). Speech bubbles were shown for 2 seconds, after which the initiating agent returned to their original position.

Given 5 × 4 = 20 possible initiator/recipient pairs, 3 possible interaction types, and 2 possible responses, the full stimulus space comprised $120^N$ possible interaction seqeunces of length $N$. To choose stimulus sequences for each study, we first randomly generated 10,000 interaction sequences. To ensure that stimuli were sufficiently informative to support meaningful inferences, we filtered out sequences that failed to meet the following criteria (see Supplementary Information for full explanation)

1. No more than two instances of the same interaction (i.e., same initiator, recipient, and response)
2. No more than one agent that is not involved in any interactions
3. No more than two and no fewer than one instances of "rejected" interactions

From the remaining sequences, we chose 8 for each study according to different study-specific criteria. For Experiment 1, we generated model predictions for each sequence, determined the 4 structures with the highest posterior probability, and then defined four target "probability profiles," choosing two sequences that matched each profile. These profiles were (see Supplementary Information for full explanation): 1 highly probable and 3 improbable structures; 1 highly probable, 1 moderately probable, and 2 improbable structures; 2 probable and 2 improbable structures; 3 moderately probable and 1 improbable structures. For Experiments 2 and 3, we chose the 8 trials with the highest sum-squared difference in prediction between the main and alternate models.

## Procedure

Participants first read a brief cover story explaining that they would watch animated videos showing social interactions between 5 people who work at the same office. Participants then read a set of instructions explaining how to interpret the speech bubble icons in the videos, and a second set of instructions showing examples of possible social structures and how to interpret them. Participants were shown the three kinds of relationships that could exist between agents (manager/subordinate, friend/friend, and mentor/mentee), and three different ways that agents could interact. Participants were told that agents will often interact in a certain way depending on their relationships, but that agents could also interact in different ways (e.g., "managers sometimes give orders to subordinates," "friends sometimes invite each other to hang out after work," "mentees sometimes ask their mentors for advice"), but were not told the actual probabilities of different interactions.

After reading the instructions, participants answered 2–4 comprehension check questions. Participants who failed at least one question were shown the instructions a second time, then given a second chance to pass the comprehension checks. Participants who failed a second time were excluded from analysis, while those who passed were shown a single example trial to familiarize themselves with the task. In Experiment 1, participants were shown 4 possible social structures in each trial, and answered "which [hierarchy/friendship groups/mentorship network] seems most likely correct" on a 7-point Likert scale from 1 (definitely not this one) to 7 (definitely this one) for each structure. In Experiment 2, participants were shown a single target agent and an interaction type, and answered which of three other agents the target was most likely to interact with, using continuous sliders ranging from 0 (definitely not this person) to 100 (definitely this person) for each agent. In Experiment 3, participants were shown a target agent and a decision they were deliberating on (working an extra shift, seeing a certain movie, taking an optional course), and answered which of the four other agents could most likely convince them to do the activity, using a 7-point Likert scale from 1 (definitely not this person) to 7 (definitely this person) for each agent. Each participant saw a random subset of four trials (out of eight total) in a random order, and randomization was balanced to ensure an even number of participants for each trial. After completing the trials, participants filled out a brief questionnaire to solicit demographic information and any feedback or concerns about the study.

## Analysis

Data analysis was performed using RStudio (version 2024.04.2+764) and R packages tidyverse (2.0.0) and boot (1.3-28.1). Across all three experiments, model fits were computed using Pearson correlations between model predictions and participant judgments (see below for data processing details). To test for statistically significant differences between models, we computed bootstrapped 95% confidence intervals over difference-in-correlation between our main model and each alternate model, and interpreted a confidence interval entirely above 0 as a significant difference (in favor of the main model).

**Experiment 1.** In each study of Experiment 1, participants rated the posterior probability of 4 different candidate structures for each of 8 trials, yielding 32 total responses. As pre-registered (https://doi.org/10.17605/OSF.IO/M75DG), we first z-scored each participant's responses across trials, then computed the average z-scored response across participants for each response variable. We then z-scored each set of model predictions within each trial (separately for each model), and computed Pearson correlations between z-scored participant responses and z-scored model predictions across all 8 trials separately for each study. To determine whether our main model outperformed alternatives by a statistically significant margin, we computed a bootstrapped 95% confidence interval of difference in correlation, and

interpreted an interval entirely above 0 as a statistically significant difference (in the main model's favor). To determine whether participants actually relied on the specific social intuitions encoded in the naive sociology models (as opposed to general statistical inference), we generated two sets of alternate predictions for each study by switching the interaction model used to compute structure probabilities (e.g., interpreting orders using the intuitions that govern invitations). We then z-scored these alternate predictions in the same fashion as the main model and computed Pearson correlations with participant responses.

**Experiment 2.** In each trial of each study, participants rated the likelihood of an initiating agent interacting with each of 3 possible recipients, yielding 24 total responses. As pre-registered, we first normalized each participant's responses for each trial to obtain a probability distribution over recipients, then averaged responses across participants within each trial and response variable. To generate model predictions, we first applied our main model to each stimulus sequence to estimate the posterior structure probability $P(S|D)$. We then computed the likelihood of a future interaction $d$, given the previously observed interactions $D$, by marginalizing over the structure posterior according to

$$P(d|D) = \sum_S P(d|S)P(S|D) \qquad (2)$$

We additionally generated predictions from an alternate "frequency-tracking" model, which estimates the likelihood of an interaction $d$ as the raw frequency with which the same interaction occurred in the stimulus sequence $D$. For interactions that were not observed in the stimulus sequence, we sampled an interaction probability from a uniform prior.

**Experiment 3.** In each trial, participants answered three questions, one for each type of social influence, judging the likelihood that each of four agents was the one who convinced a fifth target agent to take a specific action. As pre-registered, we first z-scored responses within participant, then averaged the z-scored responses across participants within each trial. To generate model predictions, we assumed that for each action type, the agent with the most influence over the decider would have a specific relation to the decider (working an extra shift-manager; seeing a movie- friend; taking a class- mentor). We therefore computed agent $i$'s influence over agent $j$'s decision as the inferred probability that agent $i$ had the corresponding relation with agent $j$. For example, if agent $j$ is considering an extra shift, then agent $i$'s degree of influence over $j$ is the likelihood that $i$ is $j$'s manager, which we computed by marginalizing over the inferred posterior $P(S|D)$, i.e.,:

$$P(i \text{ is } j\text{'s manager}|D) = \sum_S \mathbb{I}\left[i \text{ is } j\text{'s manager}|S\right]P(S|D) \qquad (3)$$

We also generated predictions from three alternate models for each trial: the first two were generated by permuting the naive sociology modules (as in Experiment 1), and the third estimated the degree of $i$'s influence over $j$ as the frequency with which $i$ initiated an interaction with $j$. We then z-scored model predictions within each trial, and computed Pearson correlations between each of our four model's predictions and participant responses.

**Reporting summary**
Further information on research design is available in the Nature Portfolio Reporting Summary linked to this article.

## Data availability
All datasets generated and analyzed for the present studies are available in an OSF repository at: https://doi.org/10.17605/OSF.IO/M75DG

## Code availability
All models were written in WebPPL, a probabilistic programming language for generative models. All model code and model predictions are available in an OSF repository at: https://doi.org/10.17605/OSF.IO/M75DG

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

## Acknowledgements

This work was supported by NSF awards IIS-2106690 (JJE) and BCS-2438827 (YD, ID, JJE)

## Author contributions

I.D., J.J.E., and Y.D. contributed to the conception, methodological design, and investigation. I.D. developed the computational models and software with guidance from J.J.E, and wrote the original manuscript with critical edits and review from J.J.E and Y.D. Y.D., J.J.E., and I.D. secured funding and resources.

## Competing interests

The authors declare no competing interests.
