## [Transparent Peer Review file · Nature Communications]

Inferring the internal structure of groups through the integration of statistical learning and causal reasoning

Corresponding Author: Dr Isaac Davis

Version 0:

Reviewer comments:

Reviewer #1

(Remarks to the Author)

This manuscript tests the hypothesis that people combine culturally derived expectations about how certain types of relationships function (e.g., a friendship is usually mutual while a mentorship relationship is not; a boss gives an employee an order) with statistical learning to infer the social structure of a group. They focused on three types of social connections: mentorship, authority, and friendship. They found that the “main” models (those adhering to the unique assumptions of each “sociology”) fit the participant data best across three experiments. I thought the manuscript was interesting and contributed new ideas to the literature examining how people learn social networks, which often focuses on one type of social relationship at a time. However, I believe there are a few points that need clarifying.

1. My main comment centers around how the models were defined and how they differed from each other. The authors state that the model “embeds a particular set of psychological assumptions about how observers form beliefs about social structure” (pp. 10-11), but I did not understand what, exactly, these assumptions were, and how they were integrated into each model. How exactly did the authors derive these models? Further explanation of the different models (explicitly stating these assumptions), especially in the main text, would help clarify this.

2. It would be helpful to know more about how the four structures in Experiment 1 were selected and explained to the participant. I struggled to understand the diagrams in Figure 2, especially as they relate to friendship (what does an arrow between friend groups mean?) and mentorship (is an arrow “asking for advice” or the inferred hierarchy in which mentors are usually higher than their mentees). If I understood correctly, the authors tried to choose structures with varying levels of likelihood given the interaction sequence. Were these likelihoods derived from the “main” model of that type of interaction only? Were the other models incorporated at all into the selection process (e.g., one that was likely for each type of sociology) in order to directly test which one the participants found most likely?

3. Looking across the three experiments, I wonder why the authors did not include the frequency model in Experiment 1, and why they did not test the two alternative models in Experiment 2. If possible, it would be helpful to see all results for each experiment, as is shown in Experiment 3.

4. There is a growing body of research (see the somewhat old now but excellent review, Brands, 2013) examining how people infer social network structures (e.g., from frequency of observed interactions) and use schemas to scaffold their learning (e.g., assuming small-worldedness). How do the authors connect this literature with their proposed naïve sociologies framework?

(Remarks on code availability)

Reviewer #2

(Remarks to the Author)

Dear Editor,

The authors investigated how humans infer the structure of social relationships through observation of social interactions (e.g., seeing work-related directives and inferring associated hierarchies). They first developed a computational model that integrates a domain-general Bayesian updating mechanism with domain-specific knowledge about social contexts—what the authors term "intuitive sociologies." This model was tested across three sets of studies in which participants observed animated videos depicting sparse social interactions among five agents. Participants were then asked to make judgments about the underlying social structure (Experiment 1), future social behaviour (Experiment 2), or social influence dynamics (Experiment 3) among the agents.

The results suggest that a) participants could accurately reconstruct internal social structures from brief observations; b) participants could use these inferred structures to predict future social interactions; and c) that participants leveraged inferred influence hierarchies to determine which agents were most likely to persuade a focal individual to engage in a particular activity. Model-based data analyses provided converging evidence that people's social inference reflects both domain-general statistical learning and domain-specific intuitions about social relationships in human groups.

In sum, this study offers compelling quantitative evidence on how humans derive latent social structures from noisy observational input—an ability that arguably constitutes one of the major adaptive challenges in human social life. The sequence of three experimental studies is logically structured and compelling. The proposed computational modelling framework represents a significant methodological advance in the study of social cognition, a field historically reliant on verbal and often imprecise conceptual models from social psychology. While the social interactions examined in this study were limited to simple, static relationships within small groups, the proposed framework is sufficiently general to be applied to more complex and dynamic social environments—an exciting avenue for future research.

We have no major concerns really, but we do include a few minor comments which we would appreciate the authors addressing in the revision. We would like to thank the authors for submitting such an excellent piece of work.

Minor Points

1. Abstract clarification

There appears to be a discrepancy between the abstract and the main text—specifically in the descriptions of Experiments 2 and 3. The abstract does not seem to clearly reflect what was actually done in these experiments. If this interpretation is correct, we suggest clarifying these descriptions for consistency.

2. Additional reference suggestion

When discussing naive sociology in infancy (page 9), the authors might also want to consider citing Kanakogi et al. (2017, *Nature Human Behaviour*):
<https://www.nature.com/articles/s41562-016-0037>

3. Inference of relationship strength

While the main analyses focused on inferring the structure of social relationships (e.g., identifying clusters of friends), we wondered whether the model or participants could also detect differences in relationship strength within those clusters (e.g., which individuals are more closely connected than others). Can the thickness of the arrows be interpreted as an index of tie strength? A brief discussion on this would be helpful.

4. Mentorship vs. authority

Are mentorship and authority simply two sides of the same coin? Figure 1 suggests that the key difference lies in the direction of interaction: in authority networks, superiors initiate actions (work orders), whereas in mentorship networks, it is the mentees who initiate actions (questions). In Figure 3, the mentorship-tuned model appears to correlate strongly negatively with data from Study 1c, and vice versa for the authority-tuned model in Study 1a. This raises the question of whether these divergent results are an artifact of modelling the difference between these networks purely in terms of arrow direction. A clarification on this point would be valuable.

Wataru Toyokawa & Jingyu Xi

(Remarks on code availability)

Reviewer #3

(Remarks to the Author)

(Remarks on code availability)

Reviewer #4

(Remarks to the Author)

This paper presents a computational model of 'naive sociology' and tests it in a series of experiments. The model takes as input observed social interactions and outputs a posterior probability distribution over social structures using Bayes' rule. A key contribution of this work is showing that people use domain-specific knowledge given by the observed interaction type (mentorship, authority, or friendship in these experiments). The model assumes that this interaction type information is available in the observed data.

Overall, I thought this was an interesting, well-written paper. It will make an important contribution to the burgeoning literature on social structure learning, which has built a bridge between social cognition and computational theories of structure learning.

Specific comments:

It was hard to understand all the modeling details, in part because some important information is relegated to the supplement. I think the model should be described in greater detail in the main text.

Clustering is an important part of the model, but it's barely discussed at all in the main text. I found that strange.

For all data plots, the caption should specify what the error bars show.

For scatter plots, what does each dot correspond to? Average responses across trials for a single participant?

There's some inconsistency in the notation. The likelihood is initially defined with $L_t(I|S)$ and then later as $P(I|S)$. I think the latter should be used.

The Chinese Restaurant Process needs to be more completely defined for readers to understand what it is and how it's being used. Also, Griffiths et al. (2003) isn't the canonical reference; typically one would cite Aldous (1985).

Please number equations.

Supplement, p. 3 & 5: "sum" -> "\sum"

(Remarks on code availability)

Version 1:

Reviewer comments:

Reviewer #1

(Remarks to the Author)

I would like to thank the authors for their thoughtful responses and edits. The authors have addressed all of my concerns.

(Remarks on code availability)

Reviewer #2

(Remarks to the Author)

Dear Editor,

Thank you for the opportunity to review the manuscript entitled "Inferring the internal structure of social collectives". We have carefully read the authors' point-by-point response to the comments raised by both us and other reviewers. The authors have addressed the reviewers' concerns thoroughly and made appropriate revisions. We are satisfied with the current version of the manuscript and believe it now meets the journal's criteria for publication. We thank the authors for their hard work and congratulations to the wonderful study.

All the best,

(Remarks on code availability)

Reviewer #3

(Remarks to the Author)

(Remarks on code availability)

Reviewer #4

(Remarks to the Author)

I am satisfied with the response of the authors to my comments.

(Remarks on code availability)

Below is a point-by-point response to each of the reviewers' comments in a separate document. To make the letter easier to read, reviewer comments appear in blue font, our responses in black font, and paper quotes appear in black font with 0.5 inch justification.

Reviewer #1

This manuscript tests the hypothesis that people combine culturally derived expectations about how certain types of relationships function (e.g., a friendship is usually mutual while a mentorship relationship is not; a boss gives an employee an order) with statistical learning to infer the social structure of a group. They focused on three types of social connections: mentorship, authority, and friendship. They found that the "main" models (those adhering to the unique assumptions of each "sociology") fit the participant data best across three experiments. I thought the manuscript was interesting and contributed new ideas to the literature examining how people learn social networks, which often focuses on one type of social relationship at a time. However, I believe there are a few points that need clarifying.

Thank you for this positive appraisal. See below for our responses to specific comments.

1. My main comment centers around how the models were defined and how they differed from each other. The authors state that the model "embeds a particular set of psychological assumptions about how observers form beliefs about social structure" (pp. 10-11), but I did not understand what, exactly, these assumptions were, and how they were integrated into each model. How exactly did the authors derive these models? Further explanation of the different models (explicitly stating these assumptions), especially in the main text, would help clarify this.

Thank you for the suggestion and for pointing out an area in which we were not sufficiently clear. At a high level, each naive sociology model encodes three sets of intuitions: who is likely to initiate a certain type of interaction, who the interactions are likely to be directed towards, and how the recipient of the interaction is likely to respond (given the identity of the initiator). Each of these intuitions depends in different ways on certain features of the underlying social structure (e.g.: the relative position in the hierarchy between initiator and recipient) as well as certain observer- or culturally- specific expectations (e.g.: the strictness of norms surrounding disagreeing with authority figures). We have made several edits to make this clearer in the paper. First, we have added a new table (Figure 2 in the revised manuscript) to summarize how each component of each naive sociology is dependent on different features of the social structure (pg 12):

Sociology	Interaction term		
	Who is the initiator?	Who is the recipient?	Does the recipient accept?
	P(init struct)	P(recip init, struct)	P(response recip, init, struct)
Authority (orders)	Agents with more subordinates more likely to initiate	Agents subordinate to initiator more likely to receive Agents closer to initiator more likely to receive	Agents subordinate to initiator more likely to accept
Friendship (invites)	Agents with more friends more likely to initiate	Agents in same clique as initiator more likely to receive	Agents in same clique as initiator more likely to accept
Mentorship (requests)	Agents with more mentors more likely to initiate Agents with more mentees less likely to initiate	Mentors of initiator & agents with same mentor more likely to receive Agents closer to initiator more likely to receive	Mentors of initiator & agents with same mentor more likely to accept

Figure 2: High-level description of the three naive sociology models used in our studies (see Supplemental Materials for equations). Each row corresponds to one of the three sociology/interaction types (from top to bottom: authority/orders, friendship/invites, mentorship/advice requests), and each column corresponds to one of the three phases of interaction represented in our model. Within each cell, we list the relevant features of the social structure and how they influence the likelihood of different interactions.

Additionally, we have added a new section (1.2: Modeling naive sociology), which provides intuitive explanations of the general assumptions that structure expectations or social interactions in each sociology and the free parameters that modulate the strength of these assumptions (pg 13-15):

1.2.1: Authority/orders

In the “authority” sociology, orders are more likely to originate from agents with more subordinates, are more likely to be directed to agents subordinate to the initiator, and interactions are more likely to occur between agents that are closer within the hierarchy (e.g.: an agent is more likely to give an order to a direct subordinate than a subordinate of a subordinate, etc). Agents are more likely to fulfill orders that originate from superiors than peers or subordinates. Two free parameters $\beta_{down}, \beta_{up} \in (0, 1)$ control the

“strictness” of the hierarchy: when $\beta_{down}=1$, agents will only give orders to subordinates

directly below them in the hierarchy; when $\beta_{down}=0$, agents are equally likely to give

orders to any subordinate, regardless of their distance within the hierarchy. β_{up} is an analogous parameter for orders that travel upwards in the hierarchy (which are inherently less likely, but still possible). The third parameter, β_{pos} , captures the “social cost” of violating a norm by rejecting an order from a legitimate superior: the higher the social cost, the less likely agents are to refuse orders from superiors.

1.2.2: Friendship/invitations

In the “friendship” sociology, invitations are more likely to originate from agents in larger friend groups (cliques), are more likely to be directed to agents in the same clique, and are more likely to be accepted by agents in the same clique. A parameter β_{init} discounts the rate at which larger friend groups affect the likelihood of agents initiating invitations: for values close to zero, the size of a friend group has little effect on who is likely to initiate an invitation; for large values, invitations are increasingly likely to originate from agents in larger friend groups. Two other parameters, $\beta_{\text{high}} > \beta_{\text{low}}$, determine the “cliquishness” of the structure: when β_{high} is much larger than β_{low} , invitations will almost always stay within the same clique, and will almost always be rejected when given to an agent from a different clique. When β_{high} is closer to β_{low} , invitations between cliques become more likely.

1.2.3: Mentorship/advice requests

In the “mentorship” sociology, advice requests are most likely to originate from agents who have mentors but no mentees, and are least likely to originate from agents with mentees but no mentors. A free parameter $\beta_{\text{request}} \in (0, 1)$ affects the strength of this difference: at $\beta_{\text{request}} \approx 1$, all agents are equally likely to request advice; at $\beta_{\text{request}} \approx 0$,

advice requests will only originate from agents with mentors but no mentees. This is similar to a “reverse” version of authority, such that interactions are more likely to be initiated by agents lower in the structure and targeted towards agents higher in the structure (as opposed to orders, which flow from high-authority to low-authority agents). Unlike the authority model, however, agents in a mentorship structure may seek advice from both mentors and peers (i.e.: another agent with the same mentor as the initiator), with a parameter $\beta_{\text{recip}} \in (0, 1)$ determining the strength of the preference for seeking

advice from mentors versus peers. Finally, a “social cost” parameter $\beta_{\text{pos}} \in (0, 1)$ determines the likelihood of an agent rejecting a request from a mentee or peer.

We hope these changes alleviate the lack of clarity surrounding the model, but please let us know if there are any parts where the model description still feels unclear. We want the model to be accessible to a broad audience and appreciate your help with helping us identify that some components were under-explained. If there are additional ways we can enhance clarity we would be delighted to do further work here.

2. It would be helpful to know more about how the four structures in Experiment 1 were selected and explained to the participant. I struggled to understand the diagrams in Figure 2, especially as they relate to friendship (what does an arrow between friend groups mean?) and mentorship (is an arrow “asking for advice” or the inferred hierarchy in which mentors are usually higher than their mentees). If I understood correctly, the authors tried to choose structures with varying levels of likelihood given the interaction

sequence. Were these likelihoods derived from the “main” model of that type of interaction only? Were the other models incorporated at all into the selection process (e.g., one that was likely for each type of sociology) in order to directly test which one the participants found most likely?

Thank you for the opportunity to clarify these issues. You are correct that, for Experiment 1, the four candidate structures in each trial were selected to ensure varying levels of posterior probability (very likely, somewhat likely, somewhat unlikely, etc.) given the interaction sequence. These posterior probabilities were computed using only the main model. While we did not include the other models in the selection process, we found that this approach was sufficient to be able to tell the models apart, given that the models make very different predictions. We have made several edits to clarify this.

First, we realized that the caption in Figure 2 (now Figure 3 in the revised manuscript) did not specify that the 4 candidate structures shown were from Study 1a, and thus depicted hierarchies, where arrows go from superiors to subordinates. We have edited the caption to the figure to clarify that. In addition, we have added text to the Figure 1 caption to clarify what the arrows mean in each context (pg 9):

Examples of social structure representations and the generative models associated with each structure. Panel a) shows three social structure graphs, each with the same clusters but encoding a different social dynamic. From left to right, these are: mentorship, where arrows go from mentors to mentees; authority, where arrows go from superiors to subordinates; and friendship, where arrows are self-directed to indicate cliques.

We also added text to clarify that participants were instructed how to interpret the structure diagrams, including the relations encoded by the arrows, and the animated icons used to indicate different interaction types (pg 18):

Participants first read a set of instructions explaining how to interpret the structure diagrams, the relations encoded by the arrows, and the animated interactions between agents. Participants were then given two chances to correctly answer three comprehension check questions, ensuring a correct interpretation of instructions. Participants who correctly answered all three comprehension checks then watched a series of animated videos, each depicting 8 social interactions between 5 agents.

Finally, we added text to clarify how we chose the interaction sequences and candidate structures for each trial. The alternate models were not involved in trial selection: rather, we chose trials based on main model predictions, to ensure we included a range of trial types, including some trials where the main model predicted a single, unambiguous structure, and others where the main model predicted multiple plausible structures. This allowed us to test whether the main model could capture both qualitative patterns in participant judgments (which structures are clearly the best explanations) as well quantitative patterns in participants’ uncertainty about multiple plausible structures (pg 19):

To select the stimulus animations and candidate social structures for each trial, we first generated a large number of randomly generated interaction sequences, applied our main model to each sequence, and identified the four most likely structures according to

the main model. We then chose a set of trials (interaction sequences plus candidate structures) to ensure a range of “degrees of certainty” across trials, including some trials where the main model infers one structure unambiguously, and some trials where the main model identified multiple plausible structures with different degrees of certainty (See Methods for more details).

3. Looking across the three experiments, I wonder why the authors did not include the frequency model in Experiment 1, and why they did not test the two alternative models in Experiment 2. If possible, it would be helpful to see all results for each experiment, as is shown in Experiment 3.

Thank you for this inquiry. For the “frequency” model, we omitted it from Experiment 1 by design: Experiment 1 requires the model to output an explicit structure representation, while Experiments 2 & 3 leverage structure representations implicitly, in the service of other predictions. The frequency model, however, is meant to test the alternate hypothesis that participants may be bypassing the structure inferences entirely and simply predicting behavior directly based on surface statistics. Thus, by design, the frequency model does not actually output an inferred social structure, and cannot be used for Experiment 1, where the task requires the generation of an explicit social structure representation. We have added the following footnote to clarify this point (pg 19):

In Experiments 2 & 3, we additionally used a non-representational alternate model that simply tracks interaction frequencies between agents. Because this model does not output any explicit structure representations, instead making behavioral predictions from interaction frequencies directly, it cannot be used as an alternate model for Experiment 1, which requires the model to output explicit structure representations

We also agree that it would be useful to include results from the “alternate sociology” models in Experiment 2 as well as in Experiment 1, and have edited the figure from Experiment 2 to include them (Figure 6 in the revised manuscript). These additional results provide further support for our account, showing the importance of the specific sociology through which interactions are interpreted, but also reveal how different aspects of the sociologies can be more or less critical depending on the inference task. In addition to revising the scatterplots in Figure 6, we have added two paragraphs of text that addresses these insights, as well as a new Figure 8 showing an example trial that illustrates a key difference between the mentorship and authority sociologies in the action prediction task. The specific edits are listed below.

The newly edited results chart (Figure 6 in the revised manuscript, pg 24):

Figure 6: Summary of results from Experiment 2, showing participant judgments (y-axis) against model predictions (x-axis); each row corresponds to a sub-study (one for each interaction type), while columns correspond to models (from left to right: authority, friendship, mentorship, and the pure frequency-tracking model). Plots highlighted in color along the diagonal indicate the main model for each study.

The additional text addressing the “alternate sociology model” results (pg 27-28):

The results from the “alternate sociology” models further suggest that participants are leveraging different social expectations when predicting different types of behavior, but also reveal that different aspects of these social expectations can be more or less critical depending on the type of inference being performed. In particular, the most pronounced differences occurred when interactions typically associated with symmetrical relations (i.e., friendship) were interpreted as anti-symmetrical relations (i.e., authority and mentorship), and vice versa. In Study 2a (authority), the “friendship” model performed significantly worse than the main model ($R=.45$, 95%CI (.05, .72)). However, while the

``mentorship" model did show a lower correlation than the main model ($R=.65$, 95%CI (.34, .84)), a bootstrapped confidence interval over difference in correlation revealed that this difference was not statistically significant (95% CI: (-0.14, .37)). The results from Study 2c (mentorship) showed a similar pattern: the ``friendship" model performed significantly worse than the main model ($R=.29$, 95%CI (-.13, .62)), but the ``authority" model was not significantly different from the main model ($R=.85$, 95%CI (.67, .93)). Thus, unlike the structure inference results from Experiment 1, where the mentorship and authority models produced strongly anti-correlated predictions, these results show that the presumed directionality of the relation is much less important when predicting future behavior (though distinguishing symmetrical from anti-symmetrical relations is still quite critical).

There are several insights raised by these results. First, there is a broad sense in which mentorship and authority carry similar social expectations, as both fundamentally involve hierarchies: one based on power (authority) and one based on knowledge (mentorship). Thus, even though the mentorship and authority sociologies are not perfect mirrors of each other (as orders are strictly expected to flow down the hierarchy, while advice requests may either flow up to mentors or laterally to peers), the subtle conceptual differences between them were not sufficient to produce strong differences in action predictions. This was further exacerbated by the fact that the trials in this experiment were selected to maximally distinguish between the main sociology model for each study and the non-representational ``frequency" model, but not the alternate sociology models (which were well-distinguished in Experiment 1). There were, however, several individual trials in which the conceptual differences between authority and mentorship were apparent in both participants' and the models' predictions, one of which is highlighted in Figure 8. Thus, although the aggregate results show significant similarities between the mentorship and authority model predictions, the differing expectations about interactions between peers is still reflected in trial-level results.

The new figure highlighting a trial from Study 2a where the difference between the authority model and mentorship model is more apparent (Figure 8 in the revised manuscript, pg 29)

[Figure Redacted]

Example trial from Study 2a, highlighting a case where the subtle differences between authority and mentorship are more apparent. In this trial, both the Pink and Blue agents are seen giving orders to Yellow, suggesting that Pink and Blue are both above Yellow in the office hierarchy. Thus, when Yellow is out of the office, both participants and the authority model predict that Pink is most likely to give an order to Green or Grey, both of whom have received orders from Yellow and are therefore further down in the hierarchy. Under the mentorship model, however, where we interpret the interactions as advice requests, the fact that Pink and Blue both received advice from Yellow suggests that Pink and Blue are in the same mentorship group. Because the mentorship model predicts that agents may seek advice from both mentors and peers, this model predicts that, when Yellow is out of the office, Pink is most likely to interact with Blue. This is contrary to both the authority model and participant judgments, which predict that Pink is least likely to interact with Blue.

4. There is a growing body of research (see the somewhat old now but excellent review, Brands, 2013) examining how people infer social network structures (e.g., from frequency of observed interactions) and use schemas to scaffold their learning (e.g., assuming small-worldedness). How do the authors connect this literature with their proposed naïve sociologies framework?

Thank you for highlighting this connection. In one sense, our approach is conceptually similar to the notion of cognitive schemas in social structure inference, as we posit a set of basic expectations and simplified representations through which observers interpret social data. Where our approach diverges from this framework, however, is in the assumed specificity and flexibility of the schemas. Traditionally, social schemas are thought of as specific structure representations tied to specific heuristics or expectations (e.g.: a linear-order hierarchy tied to the expectation that everyone in the hierarchy falls into a single “pecking order”). Under this view, observers interpret social data by attempting to map their observations onto one of a fixed set of possible schemas. Our approach differs in our assumption that people can flexibly compose these basic representations to construct specific complex structures that don’t map onto any one particular prior structure. For example, one might recognize that a high-level corporate hierarchy (e.g.: management oversees both sales and research) also contains several internal hierarchies (within each of sales and research), which may also relate to each other in non-hierarchical ways (e.g.: some salespeople are in shared friend groups with some researchers). We propose that this additional flexibility is essential for navigating messy, real-world social environments, where a social context might involve multiple overlapping social structures, some of which may be more transient or permanent than others.

We have added the following text and references to the introduction to ground our framework in the existing literature on cognitive schemas and make this connection more explicit (pg 4-6):

Our theoretical proposal is conceptually related to prior work on social network cognition suggesting that people infer social structures through the lens of cognitive “schemas” (Baldwin, 1992; Brands, 2013; Freeman, 1992). These schemas provide simplified

heuristic representations of common social network structures, which shape the way people perceive and remember collections of dyadic relations between individuals (often in a way that leads to inaccurate representations of network structure, e.g.: Breza et al. 2018; Freeman and Webster 1994). Traditionally, these social schemas are thought of as simplified representations tied to specific heuristics or expectations. For example, one schema might represent friendship cliques, tied to a transitivity heuristic entailing that two people who share a mutual friend are likely also friends (Basyouni and Parkinson, 2022; Brashears and Quintane, 2015); another schema might represent hierarchies, tied to a linear-order heuristic entailing that everyone in the group falls into a single “pecking order” (Walker, 1976). Our approach diverges in the assumed specificity and flexibility of these schemas: rather than interpreting social data through one of a fixed set of specific schemas, we posit that people can flexibly compose these basic representations to reason about more complex structures that don’t necessarily map onto one existing schema. For example, we might recognize that a high-level corporate hierarchy (e.g.: management oversees both sales and research) also contains several internal hierarchies (e.g.: within each of sales and research), which may also overlap with other non-hierarchical structures in the group (e.g.: some salespeople are in shared friend groups with some researchers). To achieve these flexible inferences in a tractable way, we propose that this hallmark of human social intelligence involves the integration of two capacities. The first is a domain-general statistical learning mechanism over a space of abstract data structures (e.g.: categories, hierarchies, networks, etc.), which can be composed into representations of arbitrary complexity (Austerweil and Griffiths, 2013; Gershman and Blei, 2012). The second is a domain-specific intuitive theory of social structures (Mahalingam, 2007; Shutts and Kalish, 2021), which encodes expectations about the kinds of relations that exist between social agents (e.g.: authority, friendship), how these relations are organized within social collectives (e.g.: power hierarchies, friendship cliques), and how they influence social interactions (e.g.: giving orders, sharing compliments). Critically, these intuitive social expectations can themselves be composed along with the corresponding structure representations: for example, the basic expectations associated with a friendship clique can be composed with the basic expectations associated with a hierarchy to obtain a set of expectations governing a hierarchy within a friendship clique. Thus, our framework proposes a richer and more flexible representation space than how cognitive schemas are traditionally thought of in social network cognition). That said, we do not deny that schemas also contribute to the forms of social cognition we investigate here. More precisely specifying their unique contribution alongside the more compositional processes we propose here will be an important topic for future work.

Reviewer #2

The authors investigated how humans infer the structure of social relationships through observation of social interactions (e.g., seeing work-related directives and inferring associated hierarchies). They first developed a computational model that integrates a domain-general Bayesian updating mechanism with

domain-specific knowledge about social contexts—what the authors term "intuitive sociologies." This model was tested across three sets of studies in which participants observed animated videos depicting sparse social interactions among five agents. Participants were then asked to make judgments about the underlying social structure (Experiment 1), future social behaviour (Experiment 2), or social influence dynamics (Experiment 3) among the agents.

The results suggest that a) participants could accurately reconstruct internal social structures from brief observations; b) participants could use these inferred structures to predict future social interactions; and c) that participants leveraged inferred influence hierarchies to determine which agents were most likely to persuade a focal individual to engage in a particular activity. Model-based data analyses provided converging evidence that people's social inference reflects both domain-general statistical learning and domain-specific intuitions about social relationships in human groups.

In sum, this study offers compelling quantitative evidence on how humans derive latent social structures from noisy observational input—an ability that arguably constitutes one of the major adaptive challenges in human social life. The sequence of three experimental studies is logically structured and compelling. The proposed computational modelling framework represents a significant methodological advance in the study of social cognition, a field historically reliant on verbal and often imprecise conceptual models from social psychology. While the social interactions examined in this study were limited to simple, static relationships within small groups, the proposed framework is sufficiently general to be applied to more complex and dynamic social environments—an exciting avenue for future research.

We have no major concerns really, but we do include a few minor comments which we would appreciate the authors addressing in the revision. We would like to thank the authors for submitting such an excellent piece of work.

Thank you, we are grateful for your positive appraisal and valuable feedback.

1. Abstract clarification

There appears to be a discrepancy between the abstract and the main text—specifically in the descriptions of Experiments 2 and 3. The abstract does not seem to clearly reflect what was actually done in these experiments. If this interpretation is correct, we suggest clarifying these descriptions for consistency.

Thank you for highlighting these discrepancies. We have edited the text in the abstract and Experiments section to ensure a consistent description of Experiments 2 & 3. In the abstract, we write:

We test our account across three sets of behavioral experiments in which participants watch brief, abstract videos of dyadic social interactions and make inferences about the underlying social structures (Exp 1), predict how social behavior will unfold after a change in the structure (Exp 2), and predict the spread of social influence within the structure (Exp 3).

In Section 1.3 (Experiments), we write (pg 15):

To this end, we ran three sets of studies in which participants observed short, animated videos of social interactions between a handful of agents, then made judgments about underlying social structures (Experiment 1), future social behavior following a change to the structure (Experiment 2), or the spread of social influence (Experiment 3) among agents in the animation.

2. Additional reference suggestion

When discussing naive sociology in infancy (page 9), the authors might also want to consider citing Kanakogi et al. (2017, Nature Human Behaviour): <https://www.nature.com/articles/s41562-016-0037>

Thank you for highlighting this reference, we have added it to section 1.2 (Modeling naive sociology, pg 12):

While the origin of naive sociology is beyond the scope of this paper, past work provides compelling evidence that many core building blocks of naive sociology emerge as early as infancy, including a representation of dominance relations (Bugental, 2000; Kanakogi et al., 2017; Mascaro and Csibra, 2012)...

3. Inference of relationship strength

While the main analyses focused on inferring the structure of social relationships (e.g., identifying clusters of friends), we wondered whether the model or participants could also detect differences in relationship strength within those clusters (e.g., which individuals are more closely connected than others). Can the thickness of the arrows be interpreted as an index of tie strength? A brief discussion on this would be helpful.

Thank you for this suggestion, this is a very interesting idea. The current model could accommodate such inferences: given a set I of interactions and an inferred posterior $P(S|I)$ over social structures, one could marginalize this posterior to compute the probability that a specific relationship exists between any two agents (essentially by taking the fraction of possible social structures in which those two agents have that relationship). This probability could potentially be interpreted as an index of the strength of that relationship between individuals. We have added the following text to the General Discussion to highlight this as an exciting future direction (pg 36):

Even without richer stimuli, people may be able to make more graded inferences about the strength of relationships between specific agents within clusters. Such judgments could be formalized within the current framework: given a set of interactions between agents, one could marginalize the inferred posterior $P(S|I)$ over social structures to compute the probability that two specific agents have a certain relation to one another, which may serve as an index of relationship strength between the two agents. Future experimental work could explore how well the model predicts these more graded judgments among participants.

4. Mentorship vs. authority

Are mentorship and authority simply two sides of the same coin? Figure 1 suggests that the key difference lies in the direction of interaction: in authority networks, superiors initiate actions (work orders), whereas in mentorship networks, it is the mentees who initiate actions (questions). In Figure 3, the mentorship-tuned model appears to correlate strongly negatively with data from Study 1c, and vice versa for the authority-tuned model in Study 1a. This raises the question of whether these divergent results are an artifact of modelling the difference between these networks purely in terms of arrow direction. A clarification on this point would be valuable.

This is an important point which merits further reflection and clarification. In a broad sense, you are correct that authority and mentorship are both types of hierarchies (one based on power/status and one based on knowledge/expertise), and that the main difference between them is the expected direction of interaction between “higher-ups” and “lower-downs.” That said, there is a subtle difference between the two models, such that they are not perfect mirrors of each other: in particular, while orders are generally expected to flow down the hierarchy, our mentorship model assumes that people may request advice from either mentors or peers (other agents with the same mentors). This leads to a subtle but important difference between the two sociologies, beyond the direction of interaction. Additionally, we believe the results you reference from Study 1a, where the mentorship model shows a strong negative correlation with the authority-generated data, highlight an important feature of our account. Namely, these results show how the same core inference mechanism instantiated by our structure learning model can produce entirely flipped intuitions depending on the intuitive sociology through which the data are interpreted. Furthermore, the tight fit with human data shows that people’s intuitions can similarly “flip” depending on the specific set of social expectations through which they interpret social interactions. In this sense, we believe that the strong negative correlations raised above provide further evidence for our account. We have made several significant edits to address this point, in conjunction with several similar comments raised by other reviewers.

First, in response to comments from all three reviewers, we have added a new subsection 1.2 to explain the details of each naive sociology model in greater detail. The section on mentorship expands on the similarities and differences with authority (pg 14-15):

1.2.3 Mentorship/advice requests

In the “mentorship” sociology, advice requests are most likely to originate from agents who have mentors but no mentees, and are least likely to originate from agents with mentees but no mentors. A free parameter $\beta_{\text{request}} \in (0, 1)$ affects the strength of this difference: at $\beta_{\text{request}} \approx 1$, all agents are equally likely to request advice; at $\beta_{\text{request}} \approx 0$,

advice requests will only originate from agents with mentors but no mentees. This is similar to a “reverse” version of authority, such that interactions are more likely to be initiated by agents lower in the structure and targeted towards agents higher in the structure (as opposed to orders, which flow from high-authority to low-authority agents). Unlike the authority model, however, agents in a mentorship structure may seek advice

from both mentors and peers (i.e.: another agent with the same mentor as the initiator), with a parameter $\beta_{\text{recip}} \in (0, 1)$ determining the strength of the preference for seeking advice from mentors versus peers. Finally, a “social cost” parameter $\beta_{\text{pos}} \in (0, 1)$ determines the likelihood of an agent rejecting a request from a mentee or peer

Additionally, in response to comments from multiple reviewers, we have expanded the results section for Experiment 2 to include the full set of alternate sociology models (previously it just included the main model and frequency model), the results of which provide several additional insights. Unlike in the structure inference studies (Experiment 1), where the authority and mentorship models produced strongly anti-correlated predictions, the authority and mentorship predictions were strongly *positively* correlated in action-prediction studies (Experiment 2), though there were still stark differences when compared with the friendship model. These results provide several insights: first, they show how the presumed directionality of interaction can make a major difference or little difference depending on the type of inference being performed. When inferring the underlying social structures (Experiment 1), using the correct intuitions about direction of interaction is clearly critical for making accurate inferences. When predicting future behavior (Experiment 2), the presumed directionality of interaction makes little difference (so long as the assumption is consistent across interactions). However, the poor fit between the authority/mentorship model predictions and the friendship model predictions show that, even for predicting behavior, it is still critical to correctly identify the relations as being symmetrical (like friendship) or anti-symmetrical (like authority/mentorship). Unfortunately, because the trials for this study were originally selected to distinguish the main model from the non-representational “frequency” model, the trials were not, as a whole, well suited for distinguishing the authority model and mentorship model. There were, however, some individual trials where these differences become more apparent. Thus, in addition to the revised Experiment 2 scatterplots showing all 4 models, we added another figure highlighting one of these example trials. Our specific edits are listed below.

The newly edited results chart (Figure 6 in the revised manuscript, pg 24):

Figure 6: Summary of results from Experiment 2, showing participant judgments (y-axis) against model predictions (x-axis); each row corresponds to a sub-study (one for each interaction type), while columns correspond to models (from left to right: authority, friendship, mentorship, and the pure frequency-tracking model). Plots highlighted in color along the diagonal indicate the main model for each study.

The additional text addressing the “alternate sociology model” results (pg 27-28):

The results from the “alternate sociology” models further suggest that participants are leveraging different social expectations when predicting different types of behavior, but also reveal that different aspects of these social expectations can be more or less critical depending on the type of inference being performed. In particular, the most pronounced differences occurred when interactions typically associated with symmetrical relations (i.e., friendship) were interpreted as anti-symmetrical relations (i.e., authority and mentorship), and vice versa. In Study 2a (authority), the “friendship” model performed significantly worse than the main model ($R=.45$, 95%CI (.05, .72)). However, while the “mentorship” model did show a lower correlation than the main model ($R=.65$, 95%CI (.34, .84)), a bootstrapped confidence interval over difference in correlation revealed that

this difference was not statistically significant (95% CI: (-0.14, .37)). The results from Study 2c (mentorship) showed a similar pattern: the "friendship" model performed significantly worse than the main model ($R=.29$, 95%CI (-.13, .62)), but the "authority" model was not significantly different from the main model ($R=.85$, 95%CI (.67, .93)). Thus, unlike the structure inference results from Experiment 1, where the mentorship and authority models produced strongly anti-correlated predictions, these results show that the presumed directionality of the relation is much less important when predicting future behavior (though distinguishing symmetrical from anti-symmetrical relations is still critical).

There are several insights raised by these results. First, there is a broad sense in which mentorship and authority carry similar social expectations, as both fundamentally involve hierarchies: one based on power (authority) and one based on knowledge (mentorship). Thus, even though the mentorship and authority sociologies are not perfect mirrors of each other (as orders are generally expected to flow down the hierarchy, while advice requests may either flow up to mentors or laterally to peers), the subtle conceptual differences between them were not sufficient to produce strong differences in action predictions. This was further exacerbated by the fact that the trials in this experiment were selected to maximally distinguish between the main sociology model for each study and the non-representational "frequency" model, but not necessarily the alternate sociology models (which were well-distinguished in Experiment 1). There were, however, several individual trials in which the conceptual differences between authority and mentorship were apparent in both participants' and the models' predictions, one of which is highlighted in Figure 8. Thus, although the aggregate results show significant similarities between the mentorship and authority model predictions, the differing expectations about interactions between peers is still reflected in trial-level results.

The new figure highlighting a trial from Study 2a where the difference between the authority model and mentorship model is more apparent (Figure 8 in the revised manuscript, pg 29)

[Figure Redacted]

Example trial from Study 2a, highlighting a case where the subtle differences between authority and mentorship are more apparent. In this trial, both the Pink and Blue agents are seen giving orders to Yellow, suggesting that Pink and Blue are both above Yellow in

the office hierarchy. Thus, when Yellow is out of the office, both participants and the authority model predict that Pink is most likely to give an order to Green or Grey, both of whom have received orders from Yellow and are therefore further down in the hierarchy. Under the mentorship model, however, where we interpret the interactions as advice requests, the fact that Pink and Blue both received advice from Yellow suggests that Pink and Blue are in the same mentorship group. Because the mentorship model predicts that agents may seek advice from both mentors and peers, this model predicts that, when Yellow is out of the office, Pink is most likely to interact with Blue. This is contrary to both the authority model and participant judgments, which predict that Pink is least likely to interact with Blue.

Reviewer #4

This paper presents a computational model of 'naive sociology' and tests it in a series of experiments. The model takes as input observed social interactions and outputs a posterior probability distribution over social structures using Bayes' rule. A key contribution of this work is showing that people use domain-specific knowledge given by the observed interaction type (mentorship, authority, or friendship in these experiments). The model assumes that this interaction type information is available in the observed data.

Overall, I thought this was an interesting, well-written paper. It will make an important contribution to the burgeoning literature on social structure learning, which has built a bridge between social cognition and computational theories of structure learning.

Thank you for the positive appraisal, and the opportunity to improve and clarify our manuscript.

1. It was hard to understand all the modeling details, in part because some important information is relegated to the supplement. I think the model should be described in greater detail in the main text.

Thank you for this comment, which was echoed by all four reviewers. We focus on our explanation of the naive sociology models in this response, and provide more details on the clustering part of the model in our response to the next comment. At a high level, each naive sociology model encodes three sets of intuitions: who is likely to initiate a certain type of interaction, who the interactions are likely to be directed at, and how the recipient of the interaction is likely to respond (given the initiator). Each of these intuitions depends in different ways on certain features of the underlying social structure (e.g.: the relative position in the hierarchy between initiator and recipient) as well as certain observer- or culturally- specific expectations (e.g.: the strictness of norms surrounding disagreeing with authority figures). We have made several edits to make this clearer in the paper. First, we have added a new table (Figure 2 in the revised manuscript) to summarize how each component of each naive sociology is dependent on different features of the social structure (pg 12):

Sociology	Interaction term		
	Who is the initiator?	Who is the recipient?	Does the recipient accept?
	P(init struct)	P(recip init, struct)	P(response recip, init, struct)
Authority (orders)	Agents with more subordinates more likely to initiate	Agents subordinate to initiator more likely to receive Agents closer to initiator more likely to receive	Agents subordinate to initiator more likely to accept
Friendship (invites)	Agents with more friends more likely to initiate	Agents in same clique as initiator more likely to receive	Agents in same clique as initiator more likely to accept
Mentorship (requests)	Agents with more mentors more likely to initiate Agents with more mentees less likely to initiate	Mentors of initiator & agents with same mentor more likely to receive Agents closer to initiator more likely to receive	Mentors of initiator & agents with same mentor more likely to accept

Figure 2: High-level description of the three naive sociology models used in our studies (see Supplemental Materials for equations). Each row corresponds to one of the three sociology/interaction types (from top to bottom: authority/orders, friendship/invites, mentorship/advice requests), and each column corresponds to one of the three phases of interaction represented in our model. Within each cell, we list the relevant features of the social structure and how they influence the likelihood of different interactions.

Additionally, we have added an additional section (1.2: Modeling naive sociology), which provides intuitive explanations of the general assumptions that structure expectations or social interactions in each sociology and the free parameters that modulate the strength of these assumptions (pg 13-15):

1.2.1: Authority/orders

In the “authority” sociology, orders are more likely to originate from agents with more subordinates, are more likely to be directed to agents subordinate to the initiator, and interactions are more likely to occur between agents that are closer within the hierarchy (e.g.: an agent is more likely to give an order to a direct subordinate than a subordinate of a subordinate, etc). Agents are more likely to fulfill orders that originate from superiors than peers or superiors. Two free parameters $\beta_{down}, \beta_{up} \in (0, 1)$ control the

“strictness” of the hierarchy: when $\beta_{down}=1$, agents will only give orders to subordinates

directly below them in the hierarchy; when $\beta_{down}=0$, agents are equally likely to give

orders to any subordinate, regardless of their distance within the hierarchy. β_{up} is an analogous parameter for orders that travel upwards in the hierarchy (which are inherently less likely, but still possible). The third parameter, β_{pos} , captures the “social cost” of violating a norm by rejecting an order from a legitimate superior: the higher the social cost, the less likely agents are to refuse orders from superiors.

1.2.2: Friendship/invitations

In the “friendship” sociology, invitations are more likely to originate from agents in larger friend groups (cliques), are more likely to be directed to agents in the same clique, and are more likely to be accepted by agents in the same clique. A parameter β_{init} discounts the rate at which larger friend groups affect the likelihood of agents initiating invitations: for values close to zero, the size of a friend group has little effect on who is likely to initiate an invitation; for large values, invitations are increasingly likely to originate from agents in larger friend groups. Two other parameters, $\beta_{\text{high}} > \beta_{\text{low}}$, determine the “cliquishness” of the structure: when β_{high} is much larger than β_{low} , invitations will almost always stay within the same clique, and will almost always be rejected when given to an agent from a different clique. When β_{high} is closer to β_{low} , invitations between cliques become more likely.

1.2.3: Mentorship/advice requests

In the “mentorship” sociology, advice requests are most likely to originate from agents who have mentors but no mentees, and are least likely to originate from agents with mentees but no mentors. A free parameter $\beta_{\text{request}} \in (0, 1)$ affects the strength of this difference: at $\beta_{\text{request}} \approx 1$, all agents are equally likely to request advice; at $\beta_{\text{request}} \approx 0$,

advice requests will only originate from agents with mentors but no mentees. This is similar to a “reverse” version of authority, such that interactions are more likely to be initiated by agents lower in the structure and targeted towards agents higher in the structure (as opposed to orders, which flow from high-authority to low-authority agents). Unlike the authority model, however, agents in a mentorship structure may seek advice from both mentors and peers (i.e.: another agent with the same mentor as the initiator), with a parameter $\beta_{\text{recip}} \in (0, 1)$ determining the strength of the preference for seeking

advice from mentors versus peers. Finally, a “social cost” parameter $\beta_{\text{pos}} \in (0, 1)$ determines the likelihood of an agent rejecting a request from a mentee or peer.

2. Clustering is an important part of the model, but it's barely discussed at all in the main text. I found that strange.

Thank you for catching this oversight. We have added text in several places to explain the specific clustering process we employed, the Chinese Restaurant Process, as well as how we adapt it for our framework. First, in section 1.1. (Social structure inference), we have edited the last two paragraphs as follows (pg 7):

However, we propose that people go beyond tracking the statistical contingencies of interactions (although we consider mere statistical tracking as an alternative account that we rigorously test), by building structured causal models of a social collective. To this end, we hypothesized that people have a space of representational primitives that can be composed into structures of arbitrary complexity (Griffiths and Austerweil, 2008; Heckerman et al., 2006; Kemp and Tenenbaum, 2008), where social subgroups and roles are formalized as nodes, and relations between subgroups are formalized as edges. This flexibility is crucial, as there are usually a vast array of plausible latent structures, and the complexity of these structures is rarely known a priori. This representational space is instantiated as a generative grammar—i.e., a grammar of social structures—containing a set of basic representations and a set of rules for combining them into more complex representations (Crawford and Ostrom, 1995; Jara-Ettinger and Dunham, 2024). However, this flexibility also poses a serious challenge: how can an observer search through an infinite space of possible social structures to identify a small handful that are most likely, given the observed behavior?

Our approach addresses this challenge by using a non-parametric “Chinese Restaurant Process” (CRP) to efficiently navigate through this infinite search space (Aldous, 1985; Griffiths et al., 2011). Intuitively, a CRP starts by positing an extremely simple structure (e.g.: a single cluster), then gradually proposes incremental changes as new agents are either placed in an existing cluster (with probability proportional to the size of that cluster), or in a new cluster that becomes available for subsequent rounds of clustering. This allows the observer to infer the appropriate level of complexity necessary to explain the data. Prior work has used CRP-based clustering methods to model social structure inference (Gershman and Cikara, 2020; Son et al., 2021), but this work focused on clustering agents by perceived similarity over a static set of features, and did not model inter-cluster relations. Our approach modifies the standard CRP by clustering agents based on similar patterns of interaction with other agents, and adds a separate step for inferring edges between clusters to represent social relations. By incrementally composing these representational primitives via non-parametric inference, an observer can efficiently construct structures with an appropriate level of complexity for explaining a set of observations. Figure 1a shows three examples of social structure representations which share the same groupings, but vary the type of social relation encoded.

We have also added the following text to section 4.1 (Methods: Models and implementation) clarifying our use of the CRP prior (pg 38-39):

To compute $P(S)$, we use a Chinese Restaurant Process (Aldous, 1985), implemented via a stick-breaking process (Ishwaran and James, 2001) with a concentration parameter of 3, to define a prior over all possible partitions of agents into groups, and for each partition, we assume a .5 probability of an edge existing between each pair of groups. Because $P(S)$ specifies a distribution over an infinite set of structures (and is therefore intractable to compute explicitly), we approximated the distribution via a Metropolis-Hastings algorithm (Hastings, 1970) iterated for 60,000 samples after a 5,000 sample burn-in.

We hope these edits provide enough detail about the clustering aspect of our model, but are happy to provide more details if necessary.

3. For all data plots, the caption should specify what the error bars show.

Thank you for highlighting this. We have edited the captions for all scatterplots to reflect that the shaded regions indicate 95% confidence intervals around the line of best fit, and all bar plots to reflect that the error bars show 95% confidence intervals for average participant judgments.

4. For scatter plots, what does each dot correspond to? Average responses across trials for a single participant?

We have edited the captions for all scatterplots to reflect that each dot shows the average response for a particular target (Experiment 1: structure; Experiments 2 & 3: agent) for each trial.

5. There's some inconsistency in the notation. The likelihood is initially defined with $L_t(I|S)$ and then later as $P(I|S)$. I think the latter should be used.

We have replaced all instances of $L_t(I|S)$ with $P(I|S)$, and double checked to ensure consistent notation throughout the paper and supplemental materials.

6. The Chinese Restaurant Process needs to be more completely defined for readers to understand what it is and how it's being used. Also, Griffiths et al. (2003) isn't the canonical reference; typically one would cite Aldous (1985).

Thank you for bringing this to our attention. Our response to your comment #2 shows the edits we made to explain the Chinese Restaurant Process and how we modified it for our framework. We have also edited all references to the Chinese Restaurant Process to include Aldous (1985).

7. Please number equations.

We have added numbers to all equations in the main text, methods, and supplemental materials.

8. Supplement, p. 3 & 5: "sum" -> "\sum"

Thank you for catching this, we have made the appropriate corrections and double checked to make sure the error does not occur elsewhere.